# Mental illness and COVID-19 vaccination: a multinational investigation of observational & register-based data

Mary M. Barker [1,20] ✉, Kadri Kõiv [2,20], Ingibjörg Magnúsdóttir [3,20], Hannah Milbourn [4,20], Bin Wang [5,6,20], Xinkai Du[7,8,20], Gillian Murphy [1], Eva Herweijer[1], Elísabet U. Gísladóttir[1], Huiqi Li [9], Anikó Lovik [1,10], Anna K. Kähler[1,11], Archie Campbell [4], Maria Feychting [1], Arna Hauksdóttir[3], Emily E. Joyce [1], Edda Bjork Thordardottir[3], Emma M. Frans[11], Asle Hoffart [7,8], Reedik Mägi [2], Gunnar Tómasson[3,12], Kristjana Ásbjörnsdóttir [3], Jóhanna Jakobsdóttir[3], Ole A. Andreassen [13,14], Patrick F. Sullivan[11,15], Sverre Urnes Johnson[7,8], Thor Aspelund [3], Ragnhild Eek Brandlistuen[6,16], Helga Ask [5,17], Daniel L. McCartney [4], Omid V. Ebrahimi [17,18], Kelli Lehto[2], Unnur A. Valdimarsdóttir [1,3,19], Fredrik Nyberg [9] & Fang Fang [1]

Individuals with mental illness are at higher risk of severe COVID-19 outcomes. However, previous studies on the uptake of COVID-19 vaccination in this population have reported conflicting results. Using data from seven cohort studies (N = 325,298) included in the multinational COVIDMENT consortium, and the Swedish registers (N = 8,080,234), this study investigates the association between mental illness (defined using self-report measures, clinical diagnosis and prescription data) and COVID-19 vaccination uptake. Results from the COVIDMENT cohort studies were pooled using meta-analyses, the majority of which showed no significant association between mental illness and vaccination uptake. In the Swedish register study population, we observed a very small reduction in the uptake of both the first and second dose of a COVID-19 vaccine among individuals with vs. without mental illness; the reduction was however greater among those not using psychiatric medication. Here we show that uptake of the COVID-19 vaccine is generally high among individuals both with and without mental illness, however the lower levels of vaccination uptake observed among subgroups of individuals with unmedicated mental illness warrants further attention.

The coronavirus disease 2019 (COVID-19) pandemic was an unprecedented global health crisis, which, as of August 2023, had caused 6.9 million deaths globally[1]. Although multiple effective vaccines against COVID-19 were developed and distributed globally, vaccine hesitancy and refusal were observed worldwide[2-5]. Crucially, the success of vaccination programmes in controlling the COVID-19 pandemic relies on high vaccination coverage[6]. Furthermore, the risk of severe COVID-19 infection and COVID-19-related mortality has been shown to be significantly higher among certain vulnerable population groups, such as individuals with mental illness (e.g. substance use disorder and psychiatric disorders requiring psychiatric hospital admissions)[7,8]. Therefore, high coverage of COVID-

19 vaccination is especially important among these high-risk groups.

Previous systematic reviews exploring the association between mental illness and uptake of various vaccinations have reported heterogenous results[9,10]. Similarly, findings from previous nationwide studies of COVID-19 vaccination uptake in people with mental illness have been inconsistent. As such, while the majority of previous studies have demonstrated lower COVID-19 vaccine uptake among individuals with certain types of mental illness such as schizophrenia and substance use disorder, uptake has been shown to be higher among individuals with anxiety or depression, compared to those with no mental illness[11–14]. Furthermore, one study which explored associations between the use of various types of prescribed psychiatric medications and COVID-19 vaccine uptake found that while individuals using antipsychotics, anxiolytics or hypnotics had lower vaccine uptake, no significant difference in uptake was observed for individuals using antidepressants[15]. However, previous studies have not investigated the associations between mental illness severity or medication status (i.e. medicated vs. unmedicated mental illness) and COVID-19 vaccination. In order to explore this, we used data on mental health diagnoses and symptoms from cohort studies included in the multinational COVID-MENT consortium[16], in addition to diagnostic and prescription data from the nationwide Swedish registers. Our hypothesis was that individuals with mental illness would have lower uptake of COVID-19 vaccination in general, and that this association would differ by mental illness type, severity, and medication status.

## Results

### COVIDMENT study analysis

Of the 403,794 individuals included in the participating COVIDMENT cohort studies, 325,298 individuals met the eligibility criteria for the present study (Supplementary Fig. 1). Over half of the overall study population (65.1%), and of each participating cohort, were female (Supplementary Table 8). The mean age in the participating cohorts ranged from 36.9 years (MAP-19) to 59.4 years (CovidLife), with a mean of 48 years in the overall study population.

The proportion of females was higher among individuals with (72.0%) vs. without (60.9%) a diagnosis of any mental illness, while the mean age was higher among those with (48.5 [SD: 1.8] years) vs. without (47.8 [SD: 3.6] years) such diagnosis (Table 1). The proportion of individuals with a previous COVID-19 infection was similar between the two groups (2.5% and 2.3% respectively), while the proportions of individuals who smoked or had ≥1 chronic physical condition were higher among those with (21.5% smoked, 66.9% had ≥1 chronic physical condition) vs. without (17.0% smoked, 36.8% had ≥1 chronic physical condition) a diagnosis of any mental illness. Low levels of missing data were observed for the majority of covariates.

314,827 individuals were included in the analysis of uptake of the first dose of a COVID-19 vaccine by 30th September 2021 (Table 2). Overall vaccination uptake was high (85.1%; $n = 267,981/314,827$). However, a small difference in uptake was observed between individuals with (82.4%; n = 99,041/120,212) vs. without (86.8%; $n = 168,174/$ 193,706) any mental illness. Vaccination uptake in each included cohort is displayed in Supplementary Table 9.

Results from the meta-analysis showed no significant association, after adjustment for covariates, between the diagnosis of any mental illness and uptake of the first dose by 30th September 2021 in the overall study population (pooled PR: 0.99, 95% CI: 0.97–1.00]; I²: 91.7%, $p < 0.001$) or among males and females separately (Fig. 1A, Supplementary Fig. 2A, Supplementary Table 10). Although the level of heterogeneity was high, a statistically significant association between the diagnosis of any mental illness and lower uptake of the first dose was only found in cohort-specific results from EstBB-EHR (PR: 0.97, 95% CI: 0.96–0.97) and MAP-19 (PR: 0.93, 95% CI: 0.88–0.98). No associations were observed between anxiety or depressive symptoms and uptake of

the first dose by 30th September 2021 in the overall study population. However, the sex-stratified analyses showed small but significant associations between anxiety (pooled PR: 0.97, 95% CI: 0.96–0.99; I²: 0.0%, $p > 0.05$) and depressive symptoms (pooled PR: 0.98, 95% CI: 0.96–0.99; I²: 0.0%, $p > 0.05$) and lower uptake of the first dose among males, but not females (Supplementary Fig. 2A, Supplementary Table 10).

Results from sensitivity analyses which excluded cohorts using electronic health records for the definition of exposure and/or outcome, or excluded individuals with any chronic physical condition, also showed no significant difference in uptake of the first dose by 30th September 2021 between those with vs. without a diagnosis of any mental illness (Supplementary Table 11). Additionally, results from the third sensitivity analysis, which explored potential differences related to national COVID-19 mitigation strategies and vaccination policies, showed very similar patterns in the Nordic and non-Nordic country groups, with no significant association between a diagnosis of any mental illness and vaccine uptake observed in either group.

Uptake of the first dose of a COVID-19 vaccine by 18th February 2022 was analysed in 313,584 individuals. Vaccination uptake was high (88.9%; $n = 278,887/313,584$); however, a small difference in uptake remained between individuals with (86.7%; $n = 103,955/119,908$) vs. without (90.3%; $n = 174,612/193,340$) mental illness (Table 2).

Results from the meta-analysis revealed a small association, after adjustment for covariates, between the diagnosis of any mental illness and first dose uptake by 18th February 20222 in the overall COVIDMENT study population (pooled PR: 0.99, 95% CI: 0.98–0.99; I²: 80.0%, $p < 0.001$) and among females separately (pooled PR: 0.98, 95% CI: 0.98–0.99; I²: 76.7%, $p < 0.001$), but not males (Fig. 1B, Supplementary Table 10, Supplementary Fig. 2B). No association was observed between anxiety or depressive symptoms and vaccination uptake in the overall study population or among males or females seperately. However, results from the MoBa cohort showed that uptake was slightly lower among individuals with vs. without anxiety (PR: 0.97, 95% CI: 0.96–0.99) or depressive (PR: 0.98, 95% CI: 0.97–0.99) symptoms.

Results from the first two sensitivity analyses showed no significant difference in first dose uptake by 18th February 2022 among individuals with vs. without a diagnosis of any mental illness (Supplementary Table 11). Although the results for both Nordic (pooled PR: 0.99, 95% CI: 0.99–1.00; I²:18.2%, $p > 0.05$) and non-Nordic (pooled PR: 0.98, 95% CI: 0.97–0.99, I²: 67.4%, $p < 0.05$) country groups were very similar, the association between a diagnosis of any mental illness and vaccine uptake was only statistically significant in the latter group.

264,404 individuals were eligible for the analysis of uptake of the second dose of a COVID-19 vaccine by 18th February 2022 (Table 2). Among these individuals, vaccination uptake was very high (95.5%; n = 252,439/264,404) and the difference in uptake between those with (94.7%; n = 93,420/98,671) vs. without (95.9%; n = 158,830/165,542) any mental illness was very small. Due to low numbers of participants, models could not be run in the MAP-19 and CovidLife cohorts.

The meta-analysis of the remaining eligible cohorts showed no significant differences, after adjustment for covariates, in second dose uptake by the diagnosis of any mental illness or the presence of anxiety or depressive symptoms in the overall study population or among males or females separately (Fig. 1C, Supplementary Table 10, Supplementary Fig. 2C). Sensitivity analysis results also showed no significant difference in vaccination uptake among those with vs. without a mental illness diagnosis (Supplementary Table 11).

### Swedish register study analysis

Among the 8,080,234 individuals included in the Swedish register study population, individuals with a specialist diagnosis of any mental illness were more likely to be female (55.7% vs. 49.6%) and younger (45.2 years vs. 50.2 years), compared to those without such diagnosis (Table 3). Individuals with a mental illness diagnosis had a lower

**Table 1 | Distribution of sociodemographic variables in the included COVIDMENT study population, overall and by diagnosis of any mental illness diagnosis, presented as N (%) or mean [SD]**

| | Diagnosis of any mental illness | | | |
| --- | --- | --- | --- | --- |
| | Yes (n = 122,976) | No (n = 201,273) | Missing (n = 1,049) | Total (N = 325,298) |
| **Cohort** | | | | |
| EstBB-C19 (Estonia) | 3107 (2.5%) | 2526 (1.3%) | 0 (0.0%) | 5633 (1.7%) |
| EstBB-EHR (Estonia) | 95,208 (77.4%) | 88,124 (43.8%) | 0 (0.0%) | 183,332 (56.4%) |
| C-19 Resilience (Iceland) | 2895 (2.4%) | 7283 (3.6%) | 239 (22.8%) | 10,417 (3.2%) |
| MAP-19 (Norway) | 698 (0.6%) | 2722 (1.3%) | 474 (45.2%) | 3894 (1.2%) |
| MoBa (Norway) | 15,496 (12.6%) | 87,315 (43.4%) | 0 (0.0%) | 102,811 (31.6%) |
| CovidLife (Scotland) | 1230 (1.0%) | 3499 (1.7%) | 31 (2.9%) | 4760 (1.5%) |
| Omtanke2020 (Sweden) | 4342 (3.5%) | 9804 (4.9%) | 305 (29.1%) | 14,451 (4.4%) |
| **Sex** | | | | |
| Female | 88,515 (72.0%) | 122,544 (60.9%) | 753 (71.8%) | 211,812 (65.1%) |
| Male | 34,446 (28.0%) | 78,713 (39.1%) | 205 (19.5%) | 113,364 (34.9%) |
| Other | 8 (0.0%) | 6 (0.0%) | 1 (0.1%) | 15 (0.0%) |
| Missing | 7 (0.0%) | 10 (0.0%) | 90 (8.6%) | 107 (0.0%) |
| **Age group, years** | | | | |
| 18–29 | 12,826 (10.5%) | 14,670 (7.3%) | 265 (25.3%) | 27,761 (8.5%) |
| 30–39 | 24,119 (19.6%) | 30,486 (15.1%) | 197 (18.8%) | 54,802 (16.9%) |
| 40–49 | 31,864 (25.9%) | 78,311 (38.9%) | 177 (16.9%) | 110,352 (33.9%) |
| 50–59 | 24,359 (19.8%) | 44,785 (22.2%) | 184 (17.5%) | 69,328 (21.3%) |
| 60–69 | 17,228 (14.0%) | 19,053 (9.5%) | 147 (14.0%) | 36,428 (11.2%) |
| 70+ | 12,558 (10.2%) | 13,845 (6.9%) | 76 (7.2%) | 26,479 (8.1%) |
| Missing | 22 (0.0%) | 123 (0.1%) | 3 (0.3%) | 148 (0.1%) |
| Mean [SD] age, years | 48.5 [1.8] | 47.8 [3.6] | 44.1 [9.4] | 48.0 [2.9] |
| **COVID-19 infection** | | | | |
| Yes | 3094 (2.5%) | 4629 (2.3%) | 17 (1.6%) | 7740 (2.4%) |
| No | 64,035 (52.1%) | 140,877 (70.0%) | 1032 (98.4%) | 205,944 (63.3%) |
| Missing* | 55,847 (45.4%) | 55,767 (27.7%) | 0 (0.0%) | 111,614 (34.3%) |
| **Smoking status** | | | | |
| Yes | 26,465 (21.5%) | 34,253 (17.0%) | 62 (5.9%) | 60,780 (18.7%) |
| No | 86,737 (70.5%) | 155,181 (77.1%) | 493 (47.0%) | 242,411 (74.5%) |
| Missing | 9774 (8.0%) | 11,839 (5.9%) | 494 (47.1%) | 22,107 (6.8%) |
| **Chronic physical conditions** | | | | |
| 0 | 47,606 (38.7%) | 123,948 (61.6%) | 160 (15.2%) | 171,714 (52.8%) |
| 1 | 34,723 (28.2%) | 46,836 (23.3%) | 68 (6.5%) | 81,627 (25.1%) |
| 2+ | 39,497 (32.1%) | 27,288 (13.5%) | 46 (4.4%) | 66,831 (20.5%) |
| Missing | 1150 (0.9%) | 3201 (1.6%) | 775 (73.9%) | 5126 (1.6%) |

*Mainly due to COVID-19 testing data only being available for some EstBB-EHR individuals.

prevalence of university education (28.7% vs. 38.5%), were less likely cohabiting (22.3% vs. 42.5%) or in the highest quartile of income (13.0% vs. 25.8%), and had a higher prevalence of chronic physical conditions, as denoted by a CCI of ≥1 (19.0% vs. 11.0%). Although the prevalence of severe COVID-19 infection was low in the overall study population (0.4%), the prevalence was higher among individuals with (0.8%) vs. without (0.3%) any mental illness diagnosis. The proportion of missing data for all covariates was low (≤2.5%).

In this study population, uptake of the first dose of a COVID-19 vaccine by 30th September 2021 was high (84.6%; n = 6,834,074/8,080,234) (Table 4). However, vaccination uptake was slightly lower in individuals with (78.5%; n = 387,341/493,705) vs. without (85.0%; 6,446,733/7,586,529) a specialist diagnosis of any mental illness. Vaccination uptake in relation to each type of mental illness diagnosis and type of psychiatric medication used is shown in Supplementary Table 12.

Taking into account all covariates, we found that uptake of the first dose in individuals with any mental illness was 1% lower than that of individuals without a mental illness (PR: 0.99, 95% CI: 0.99–0.99, $p < 0.001$) (Fig. 2A). Similarly small differences in first dose uptake were shown for most types of mental illness, except for substance use disorder which had the strongest association with lower vaccination uptake (PR: 0.84, 95% CI: 0.84–0.85, $p < 0.001$), and depression (PR: 1.02, 95% CI: 1.02–1.02, $p < 0.001$) and bipolar disorder (PR: 1.04, 95% CI: 1.03–1.04, $p < 0.001$), for which significantly higher vaccination uptake was found. No significant associations were found for tobacco use disorder or anxiety. A 3% higher uptake was observed among individuals with prescribed use of any psychiatric medication, compared to individuals not using such medication (PR: 1.03, 95% CI: 1.03–1.03, $p < 0.001$). Similar associations were found for different types of psychiatric medication. Sex-stratified analyses revealed a significant association between the diagnosis of any mental illness and lower first dose uptake among males (PR: 0.97, 95% CI: 0.97–0.98, $p < 0.001$), but not females. The results did not differ substantially among individuals with vs. without chronic physical conditions (Supplementary Table 13).

**Table 2 | Uptake of COVID-19 vaccination overall and by diagnosis of any mental illness diagnosis, in the included COVIDMENT study population, presented as N (%)**

| | Uptake of first dose of a COVID-19 vaccine by 30th September 2021 | | | Uptake of first dose of a COVID-19 vaccine by 18th February 2022 | | | Uptake of second dose of a COVID-19 vaccine by 18th February 2022 | | |
|---|---|---|---|---|---|---|---|---|---|
| | Yes | No | Total | Yes | No | Total | Yes | No | Total |
| Total study population | 267,981 (85.1%) | 46,846 (14.9%) | 314,827 (100.0) | 278,887 (88.9%) | 34,697 (11.1%) | 313,584 (100.0%) | 252,439 (95.5%) | 11,965 (4.5%) | 264,404 (100.0%) |
| Mental illness | 99,041 (82.4%) | 21,171 (17.6%) | 120,212 (100.0%) | 103,955 (86.7%) | 15,953 (13.3%) | 119,908 (100.0%) | 93,420 (94.7%) | 5,251 (5.3%) | 98,671 (100.0%) |
| No mental illness | 168,174 (86.8%) | 25,532 (13.2%) | 193,706 (100.0%) | 174,612 (90.3%) | 18,728 (9.7%) | 193,340 (100.0%) | 158,830 (95.9%) | 6,712 (4.1%) | 165,542 (100.0%) |
| Missing | 766 (84.3%) | 143 (15.7%) | 909 (100.0%) | 320 (95.2%) | 16 (4.8%) | 336 (100.0%) | 189 (99.0%) | 2 (1.0%) | 191 (100.0%) |

Results from the multi-level exposure analysis showed that, compared to individuals with neither any specialist mental illness diagnosis nor prescribed use of any psychiatric medication, uptake of the first dose was 3% higher among those with prescribed use of any psychiatric medication but no specialist diagnosis (PR: 1.03, 95% CI: 1.03–1.03, $p < 0.001$), and 1% higher among those with both a specialist diagnosis and prescribed use of any psychiatric medication (PR: 1.01, 95% CI: 1.01–1.01, $p < 0.001$) (Fig. 3A). However, those with a specialist diagnosis of any mental illness but no prescribed use of any psychiatric medication had a 9% reduction in first dose uptake (PR: 0.91, 95% CI: 0.91–0.91, $p < 0.001$). This pattern was also observed for the multi-level exposure analysis carried out for anxiety, depression, and psychotic disorder, with a particularly low uptake of vaccination among individuals with psychotic disorder but no medication use (PR: 0.78, 95% CI: 0.76–0.79, $p < 0.001$).

6,834,054 individuals were eligible for the analysis of the uptake of the second dose of a COVID-19 vaccine by 30th November 2021 (Table 4). Vaccination uptake in the overall study population was very high (98.1%; $n = 6,704,293/6,834,054$), and the difference between those with (96.2%; $n = 372,437/387,340$) vs. without (98.2%; $n = 6,331,856/6,446,714$) a specialist diagnosis of any mental illness was very small.

Accordingly, model results showed very small differences in second dose uptake among those with vs. without a specialist diagnosis of any mental illness (PR: 0.99, 95% CI: 0.99–0.99, $p < 0.001$) or prescribed use of any psychiatric medication (Fig. 2B). This was also observed for all types of mental illness diagnosis (e.g. PR for substance use disorder: 0.94, 95% CI: 0.94–0.95, $p < 0.001$), PR for psychotic disorder: 1.00, 95% CI: 0.99–1.00, $p < 0.001$) and all types of psychiatric medication. Results from the stratified analyses showed no substantial differences in the associations by sex or the presence of chronic physical conditions (Supplementary Table 13). Simiarly, the multi-level exposure analysis showed statistically significant, but very small, differences in second dose uptake according to specialist diagnosis and/or prescribed medication use (Fig. 3B).

## Discussion

This multinational study of 325,298 individuals from the participating COVIDMENT cohort studies, and 8,080,234 individuals from the Swedish national registers, showed that uptake of COVID-19 vaccination was high, and differences in uptake by mental illness were, in general, small. The majority of the analyses conducted in the COVIDMENT study population showed no significant difference in vaccination uptake according to the presence of diagnosed mental illness or anxiety or depressive symptoms. In the Swedish register analysis, although slightly lower vaccination uptake was observed among individuals with a specialist diagnosis of a mental illness, the absolute difference in vaccination uptake was very small. We did, however, show that individuals with substance use disorder had approximately 16% lower uptake of COVID-19 vaccination, while individuals with a specialist diagnosis of mental illness without ongoing psychiatric medication (i.e., proxy of more severe illness without medical treatment) had approximately 9% lower uptake. The results were largely similar among both males and females in our study populations.

Our findings support the results of existing nationwide studies. Accordingly, previous studies have also shown significantly lower uptake of COVID-19 vaccination among individuals with substance use disorders[11,13]. Some previous studies have shown a higher uptake of COVID-19 vaccination among individuals with anxiety or depression[13,14], while another revealed an association between the use of anxiolytics, but not antidepressants, and lower COVID-19 vaccination uptake[15]. The findings from our study similarly found conflicting results, namely that results from the COVIDMENT study population generally showed no association between anxiety or depressive symptoms and COVID-19 vaccination, while results from the Swedish

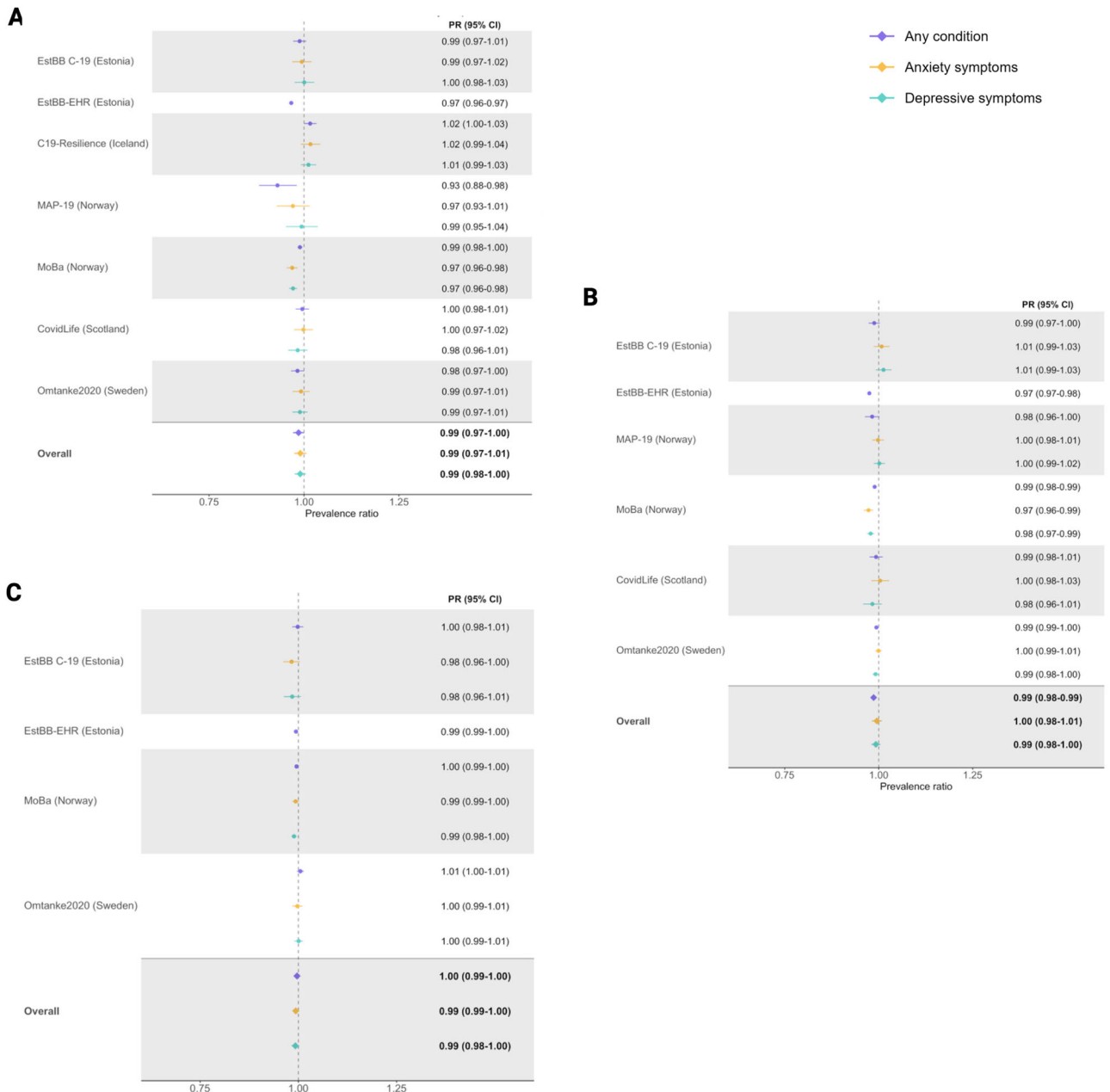

**Fig. 1 | Prevalence ratio (PR) and 95% confidence intervals (CI) of COVID-19 vaccine uptake, according to the presence of any mental illness diagnosis, anxiety symptoms or depressive symptoms, in the included COVIDMENT study population. A** first dose of a COVID-19 vaccine by 30th September 2021, (**B**) first dose of a COVID-19 vaccine by 18th February 2022, (**C**) second dose of a COVID-19 vaccine by 18th February 2022. Data are presented as PR with 95% CIs (horizontal lines), rounded to 2 decimal places. Cohort-specific estimates are adjusted for age, sex, previous COVID-19 infection, smoking, and physical comorbidity status (except MAP-19 models, which are adjusted for age, sex, and previous COVID-19 infection only). The 'overall' estimates are derived from the random effects meta-analyses of the cohort-specific estimates. EstBB cohorts (EstBB-C19 = The Estonian Biobank COVID-19 Cohort; EstBB-EHR = The Estonian Biobank electronic health records); C19-Resilience = The Icelandic COVID-19 National Resilience Cohort; MAP-19 = The Norwegian COVID-19, Mental Health and Adherence Project; MoBa = The Norwegian Mother, Father and Child Cohort Study). Total N (any mental illness diagnosis; anxiety symptoms: depressive symptoms) = (**A**) 295,319; 113,002; 110,322; (**B**) 294,647; 112,025; 108,949; (**C**) 246,043; 94,830; 92,215.

register population showed a higher first dose uptake among individuals with a specialist diagnosis of depression, but not anxiety. Furthermore, our findings show that individuals with a diagnosis of depression but not using prescribed medication had a lower uptake of the first dose of a COVID-19 vaccine. These results indicate that the association between anxiety or depression and COVID-19 vaccination uptake may differ by severity and medication status.

There are many strengths of our study. Firstly, the complementary use of the two different types of prospectively collected data sources allowed for the benefits of both the rich self-reported COVIDMENT data and the Swedish national registers, minimising concerns of selection bias, and allowing for the investigation of associations between the severity and medication status (i.e. medicated vs. unmedicated mental illness) of mental illness and COVID-19 vaccination, which has not been possible in previous studies. The multinational nature of our study also enabled us to investigate whether the results found in previous country-specific studies translate to a multinational context.

**Table 3 | Distribution of sociodemographic variables in the included Swedish register study population, overall and by specialist diagnosis of any mental illness, presented as N (%) or mean [SD]**

| | Any mental illness | | Total (N = 8,080,234) |
|---|---|---|---|
| | Yes (n = 493,705) | No (n = 7,586,529) | |
| Sex | | | |
| Male | 218,602 (44.3%) | 3,821,719 (50.4%) | 4,040,321 (50.0%) |
| Female | 275,103 (55.7%) | 3,764,810 (49.6%) | 4,039,913 (50.0%) |
| Age group, years | | | |
| 18–29 | 125,959 (25.5%) | 1,386,819 (18.3%) | 1,512,778 (18.7%) |
| 30–39 | 97,486 (19.8%) | 1,291,798 (17.0%) | 1,389,284 (17.2%) |
| 40–49 | 79,317 (16.1%) | 1,216,506 (16.0%) | 1,295,823 (16.0%) |
| 50–59 | 78,571 (15.9%) | 1,225,863 (16.2%) | 1,304,434 (16.2%) |
| 60–69 | 56,491 (11.4%) | 1,039,368 (13.7%) | 1,095,859 (13.6%) |
| 70–79 | 38,576 (7.8%) | 947,297 (12.5%) | 985,873 (12.2%) |
| 80+ | 17,305 (3.5%) | 478,878 (6.3%) | 496,183 (6.1%) |
| Mean [SD] age, years | 45.2 [18.1] | 50.2 [19.0] | 49.9 [19.0] |
| Region of residence | | | |
| East | 226,085 (45.8%) | 2,969,367 (39.1%) | 3,195,452 (39.5%) |
| South | 187,734 (38.0%) | 3,301,334 (43.5%) | 3,489,068 (43.2%) |
| North | 79,865 (16.2%) | 1,308,156 (17.3%) | 1,388,021 (17.2%) |
| Missing | 21 (0.0%) | 7672 (0.1%) | 7693 (0.1%) |
| Highest educational attainment | | | |
| University ( > 12 years) | 141,610 (28.7%) | 2,919,307 (38.5%) | 3,060,917 (37.9%) |
| Secondary school (9–12 years) | 227,668 (46.1%) | 3,255,080 (42.9%) | 3,482,748 (43.1%) |
| Primary school ( < 9 years) | 117,369 (23.8%) | 1,222,264 (16.1%) | 1,339,633 (16.6%) |
| Missing | 7058 (1.4%) | 189,878 (2.5%) | 196,936 (2.4%) |
| Cohabitation status | | | |
| Cohabiting | 109,867 (22.3%) | 3,221,005 (42.5%) | 3,330,872 (41.2%) |
| Non-cohabiting | 383,817 (77.7%) | 4,357,852 (57.4%) | 4,741,669 (58.7%) |
| Missing | 21 (0.0%) | 7672 (0.1%) | 7693 (0.1%) |
| Income | | | |
| Q1 (lowest) | 219,986 (44.6%) | 1,798,437 (23.7%) | 2,018,423 (25.0%) |
| Q2 | 121,680 (24.6%) | 1,897,728 (25.0%) | 2,019,408 (25.0%) |
| Q3 | 88,027 (17.8%) | 1,929,476 (25.4%) | 2,017,503 (25.0%) |
| Q4 (highest) | 63,991 (13.0%) | 1,953,211 (25.8%) | 2,017,202 (24.9%) |
| Missing | 21 (0.0%) | 7,677 (0.1%) | 7,698 (0.1%) |
| Severe COVID-19 infection | | | |
| Yes | 3818 (0.8%) | 24,724 (0.3%) | 28,542 (0.4%) |
| No | 489,887 (99.2%) | 7,561,805 (99.7%) | 8,051,692 (99.6%) |
| Charlson Comorbidity Index (CCI) | | | |
| 0 | 400,094 (81.0%) | 6,752,778 (89.0%) | 7,152,872 (88.5%) |
| 1 | 33,974 (6.9%) | 298,530 (3.9%) | 332,504 (4.1%) |
| 2+ | 59,637 (12.1%) | 535,221 (7.1%) | 594,858 (7.4%) |

**Table 4 | Uptake of COVID-19 vaccination overall, and by specialist diagnosis of any mental illness and prescribed use of any psychiatric medication, in the included Swedish register population, presented as N (%)**

| | First dose of a COVID-19 vaccine by 30th September 2021 | | | Second dose of a COVID-19 vaccine by 30th November 2021 | | |
|---|---|---|---|---|---|---|
| | Yes | No | Total | Yes | No | Total |
| Total study population | 6,834,074 (84.6%) | 1,246,160 (15.4%) | 8,080,234 (100.0%) | 6,704,293 (98.1%) | 129,761 (1.9%) | 6,834,054 (100.0%) |
| Any mental illness | | | | | | |
| Yes | 387,341 (78.5%) | 106,364 (21.5%) | 493,705 (100.0%) | 372,437 (96.2%) | 14,903 (3.8%) | 387,340 (100.0%) |
| No | 6,446,733 (85.0%) | 1,139,796 (15.0%) | 7,586,529 (100.0%) | 6,331,856 (98.2%) | 114,858 (1.8%) | 6,446,714 (100.0%) |
| Any psychiatric medication | | | | | | |
| Yes | 1,801,401 (87.7%) | 253,406 (12.3%) | 2,054,807 (100.0%) | 1,766,489 (98.1%) | 34,909 (1.9%) | 1,801,398 (100.0%) |
| No | 5,032,673 (83.5%) | 992,754 (16.5%) | 6,025,427 (100.0%) | 4,937,804 (98.1%) | 94,852 (1.9%) | 5,032,656 (100.0%) |

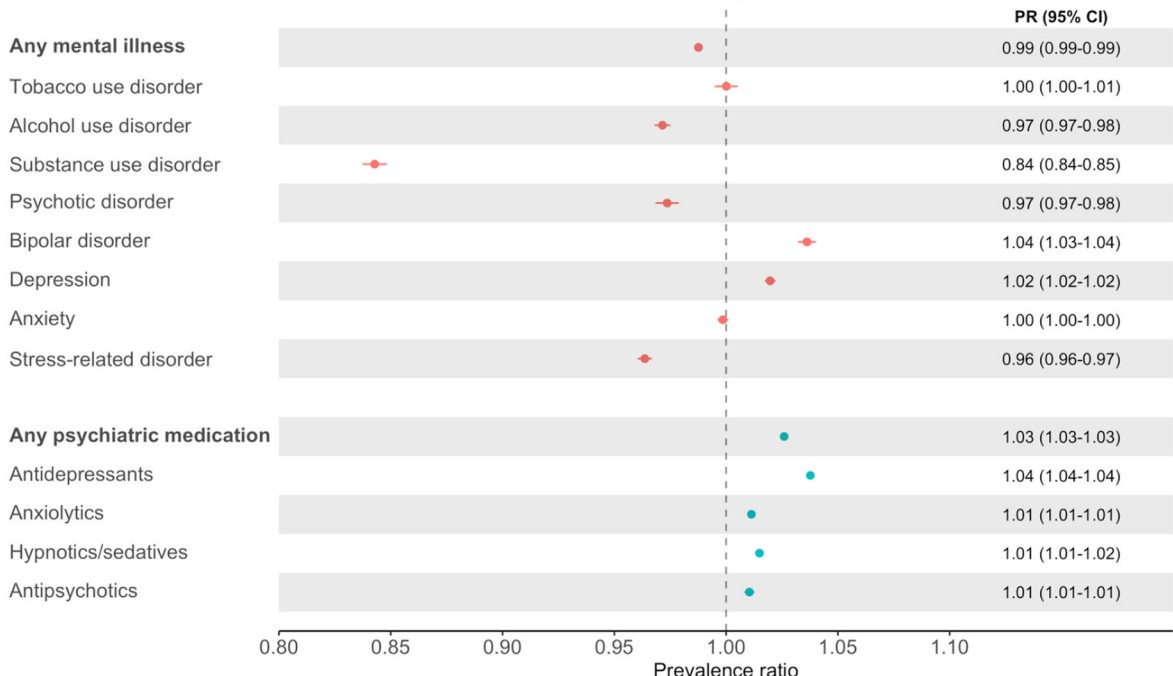

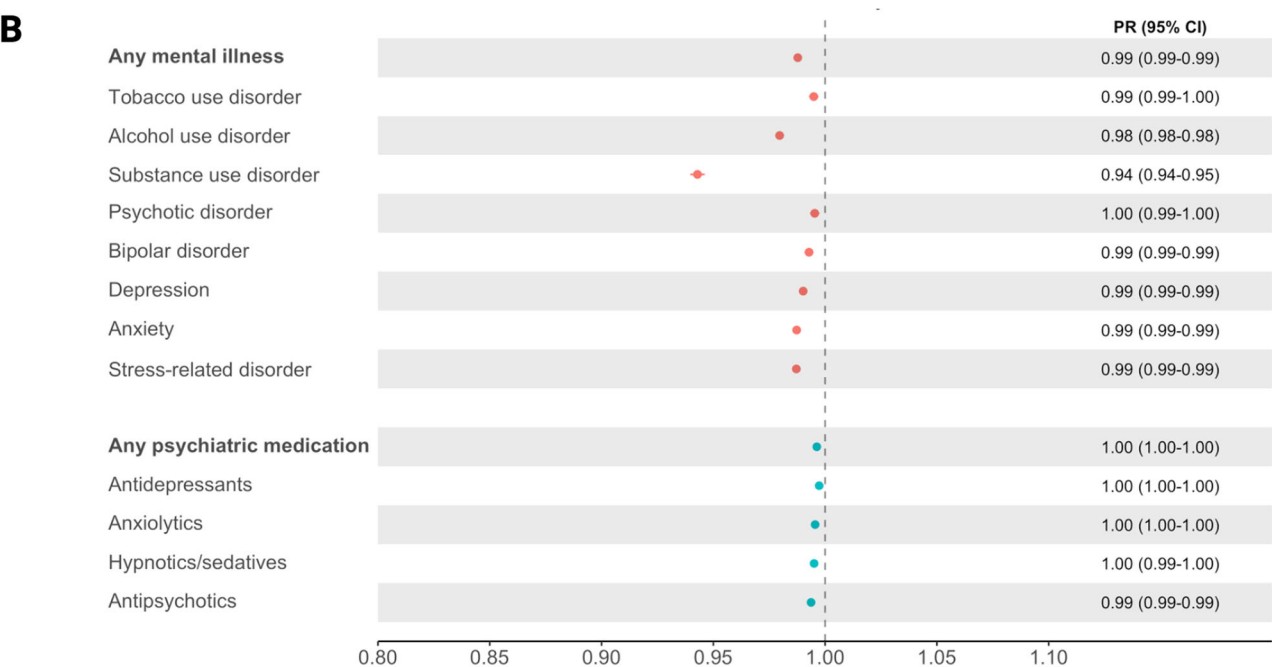

**Fig. 2 | Prevalence ratio (PR) and 95% confidence intervals (CI) of COVID-19 vaccine uptake, according to specialist diagnosis of mental illness and prescribed use of psychiatric medication, in the Swedish register study population. A** first dose of a COVID-19 vaccine by 30th September 2021 and (**B**) second dose of a COVID-19 vaccine by 30th November 2021. Data are presented as PR with 95% CIs (horizontal lines), rounded to 2 decimal places. All estimates are adjusted for age, sex, region of residence, highest educational attainment, cohabitation status, income, severe COVID-19 infection and the Charlson Comorbidity Index (CCI). Substance use disorder excludes alcohol and tobacco use disorders. N = (A) 7,883,298; (B) 6,728,266.

However, limitations of the study must also be noted. Firstly, selection bias could have been present in the COVIDMENT cohorts, whereby participants could have been less likely to have mental illness and likely to be vaccinated against COVID-19. However, by comparing the findings between the COVIDMENT study population and the Swedish register population, which would not have suffered from selection bias, we were able to investigate whether the potential selection bias in the COVIDMENT cohorts had a substantial effect on the study results. Similar findings from both data sources highlighted the robustness of the study results, even in the potential presence of selection bias.

Another potential limitation is the different multivariable adjustments used in the country-specific analysis, due to data unavailability

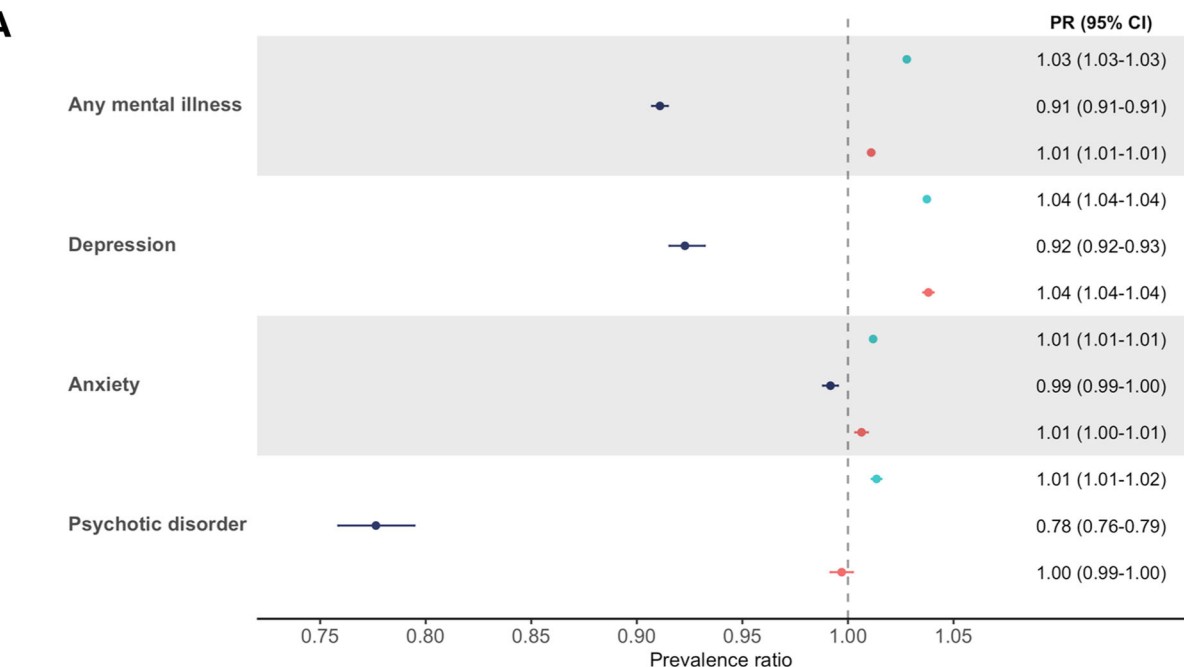

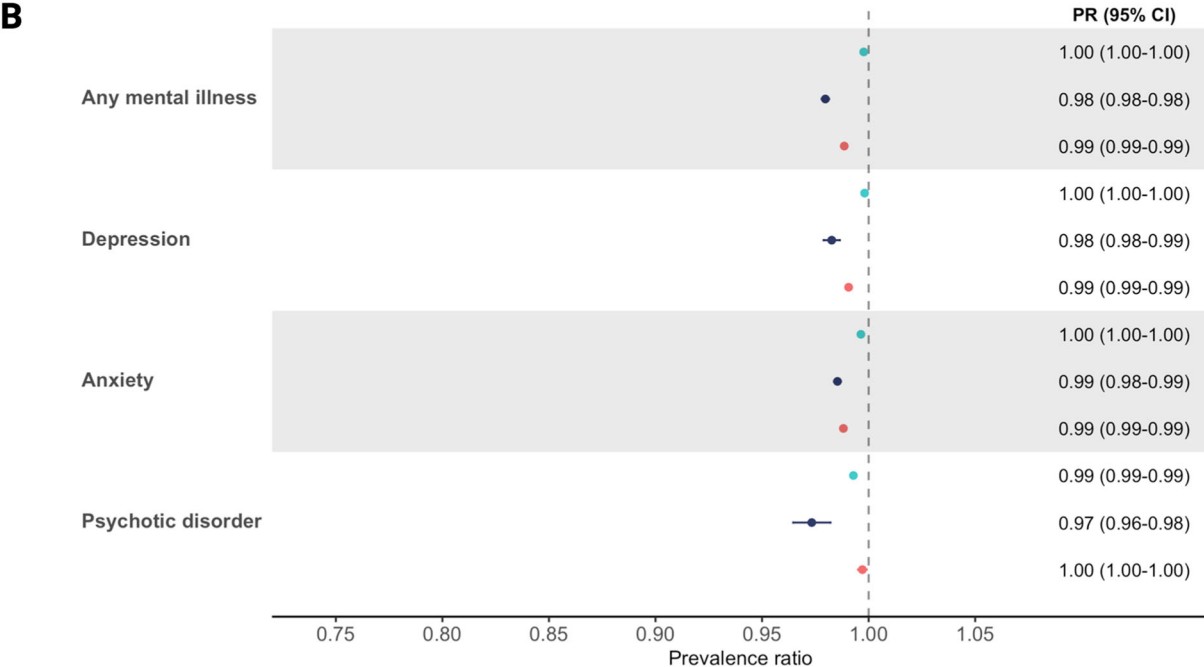

**Exposure category**

- (-) Diagnosis, (+) Medication
- (+) Diagnosis, (-) Medication
- (+) Diagnosis, (+) Medication

**Fig. 3 | Prevalence ratio (PR) and 95% confidence intervals (CI) of COVID-19 vaccine uptake, according to specialist diagnosis of mental illness/prescribed medication status, in the Swedish register study population. A** first dose of a COVID-19 vaccine by 30th September 2021 and (**B**) second dose of a COVID-19 vaccine by 30th November 2021. Data are presented as PR with 95% CIs (horizontal lines), rounded to 2 decimal places. All estimates are adjusted for age, sex, region of residence, highest educational attainment, cohabitation status, income, severe COVID-19 infection and the Charlson Comorbidity Index (CCI). N = (**A**) 7,883,298; (**B**) 6,728,266.

in some cohorts. This, in addition to the presence of differing levels of missing data, may have influenced the pooling of country-specific results and increased the heterogeneity in the meta-analyses. Another limitation is the potential for residual confounding. As some of the included COVIDMENT studies did not collect data on several socio-demographic factors such as income and educational level, we were unable to adjust for these potential confounders in the COVIDMENT analysis. However, as rich sociodemographic information is included in the Swedish registers, we were able to adjust for these potential confounders in the register-based analysis, finding similar results to those observed in the COVIDMENT cohort analysis, showing that the results were relatively robust even in the presence of these con-founders. However, as in all observational studies, there is still the possibility for residual confounding due to factors not adjusted for, one such factor being engagement with healthcare services. As such, we were only able to identify mental illness among individuals who presented to healthcare services.

Additionally, identification of mental illness in the Swedish regis-ter data was based on specialist diagnosis or prescribed use of psy-chiatric medication, which may occur some time after the onset of symptoms. Furthermore, in our multi-level exposure analysis we used diagnosis and prescription data to give indications of the severity level and medication status of mental illness. We used prescribed use of psychiatric medication as a proxy for treatment of mental illness; however, not all individuals with mental illness are in need of treatment with psychiatric medications. As the diagnostic codes used to identify mental illness in this study were from specialist care, we speculated that a substantial proportion of individuals with these diagnostic codes had relatively severe mental illness which would require some kind of treatment. However, further studies should explore patterns of treat-ment use including psychotherapy and other non-medical treatments. Another consideration is the use of different methods for the identi-fication of exposure (mental illness) or outcome (COVID-19 vaccina-tion uptake) by the COVIDMENT studies, with some using self-report measures and others using electronic health records. However, results from our sensitivity analysis in which we excluded cohorts that used electronic health records to define exposures and/or outcomes showed very similar results. Lastly, the nature of this observational study means that it is not possible to ascertain causality based on the findings.

Although the multinational nature of the study increases the representativeness of the findings, all participating countries have established welfare systems and generally accessible healthcare, meaning that caution should be taken when generalising the results to other global regions. We also observed slight differences between the cohorts, which could have been, at least partially, due to the varying prioritisation schedules for COVID-19 vaccination used in their respective countries. Although international recommendations, such as those from the European Union (EU), suggested vaccine prioritisa-tion for indivdiuals at the highest risk for severe COVID-19, countries could decide which population groups to include in their prioritisation schedules. As a result, some countries, such as Estonia, did not select individuals with mental illness for priority vaccination whereas other countries, such as Scotland, did[17–19]. Additional country-level variables, such as the specific COVID-19 mitigation and vaccination policies used may have also led to between-country differences in vaccination, by facilitating or hindering vaccine uptake in particular population groups. However, the relatively similar results observed from our sensitivity analysis, which categorised countries based on national COVID-19 mitigation and vaccination policies, suggest that these fac-tors may not have a large impact on the association between mental illness and COVID-19 vaccination.

Our findings of substantially lower vaccination uptake among individuals with unmedicated diagnosed mental illness in the Swedish register study analysis have important implications. Although we were unable to investigate the underlying reasons in the present study, lower vaccination uptake could have been due to particularly low levels of engagement with preventative healthcare in these groups[20]. Individuals with mental illness have been shown to have poor access to nonpsychiatric healthcare, including preventative services such as vaccination programmes, primarily due to barriers such as low levels of knowledge and awareness of such services, and accessibility issues[21]. However, pilot interventions aimed at increasing vaccination uptake among individuals with mental illness by addressing these barriers, for example through targeted education campaigns and the integration of psychiatric providers in vaccination programmes, have been shown to be effective[21,22]. Therefore, strategies such as these could be incorpo-rated into future vaccination campaings in order to reduce barriers to vaccination among individuals with relatively severe mental illness (e.g., attended by specialist care).

In conclusion, in this large, multinational study we showed that uptake of the COVID-19 vaccine was high, even among most indivi-duals with a mental illness, highlighting the comprehensiveness and success of the COVID-19 vaccination campaign in reaching most population groups. However, specific groups of people with a recent specialist diagnosis of mental illness yet not on psychiatric medication, were still at risk for low vaccination uptake. These findings have important implications for the design of current and future vaccina-tion campaigns against infectious diseases and future pandemics.

## Methods
### COVIDMENT study analysis
**Study population.** In order to explore the association between mental illness and uptake of COVID-19 vaccination across several countries, we leveraged data from cohort studies included in the COVIDMENT consortium, a multinational research collaboration aimed at studying mental health trajectories associated with COVID-19[16]. The COVID-MENT study population in the present study was comprised of data from seven COVIDMENT cohorts, namely the Estonian Biobank (EstBB) cohorts: EstBB full cohort with electronic health record linkages (EstBB-EHR) and the EstBB COVID-19 subcohort (EstBB-C19); the Ice-landic COVID-19 National Resilience Cohort (C-19 Resilience); the Norwegian COVID-19 Mental Health and Adherence study (MAP-19); the Norwegian Mother, Father, and Child Cohort study (MoBa)[23]; the Scottish CovidLife study; and the Swedish Omtanke2020 study. Ethical approvals for all included cohorts, encompassing the analyses con-ducted in this study, were obtained from regional or national ethics committees, and all participants provided informed consent (Supple-mentary Table 1). The cohorts used different recruitment strategies, with most recruiting from established cohorts whilst others also allowed self-recruitment via social media. The number of data collec-tion waves varied between the cohorts, with the first data collected in March 2020.

**Exposure variables.** All exposure variables referred to mental illness experienced before the initiation of COVID-19 vaccination in each study country (Supplementary Table 2). The primary exposure variable was the lifetime diagnosis of any mental illness, which, for all cohorts except EstBB-C19 and EstBB-EHR, was defined using self-report data from questionnaire items asking participants if they had ever been diagnosed with any mental illness. In EstBB-C19 and EstBB-EHR, ICD-10 codes were used to define mental illness diagnoses through linked EHRs (containing diagnoses from Estonian Health Insurance Fund (HIF) treatment bills since 2004), including primary/secondary care and inpatient/outpatient diagnoses (Supplementary Table 3). All included cohort studies also collected data on anxiety and depressive symptoms, therefore these data were included as secondary exposure variables. In the majority of cohorts, moderate-to-severe anxiety and depressive symptoms, during the past two weeks, were defined as total scores of ≥10 from the Generalised Anxiety Disorder Assessment (GAD-

7) and Patient Health Questionnaire-9 (PHQ-9) scales, respectively[24,25]. In EstBB-C19, anxiety and depressive symptoms, during the past four weeks, were measured using the Emotional State Questionnaire (EST-Q2), utilising cut-off values of >11 to define moderate-to-severe anxiety or depressive symptoms[26].

**Outcome variables.** The primary outcome variable was (i) first dose of a COVID-19 vaccine by 30th September 2021. This end date represents the time at which all included countries had offered COVID-19 vaccination to all adult residents and cohort participants had adequate opportunity to book and attend vaccination appointments (Supplementary Table 2). Two additional outcome variables were used: (ii) first dose of a COVID-19 vaccine by 18th February 2022, and (iii) second dose of a COVID-19 vaccine by 18th February 2022. This second date represents the time point at which the majority of cohorts had conducted a new wave of data collection, and therefore it was the first date post-September 2021 at which vaccination uptake could be re-assessed. As C-19 Resilience did not conduct further data collection between September 2021 and February 2022, it was only included in the analysis of outcome (i). Participants were eligible for the present study if they had available data for outcomes (i) and/or (ii). Participants were included in the analysis of outcome (iii) if they had received the first dose of any COVID-19 vaccine, except the JCOVDEN vaccine (for which a one-dose schedule was used[27]), before 18th February 2022. In C-19 Resilience, MAP-19, MoBa, and Omtanke2020, vaccination uptake was defined using self-report data collected in various cohort-specific follow-up questionnaires. The remaining cohorts used linked EHR data to define vaccination uptake.

**Covariates.** Potential confounders known, based on literature and biological relevance, to be associated with both the exposure (mental illness) and outcome (COVID-19 vaccination uptake) were included as covariates. All covariates (age, sex, smoking status, previous COVID-19 infection, and physical comorbidity status) were defined using data collected before COVID-19 vaccination was initiated in each study country. Smoking was used as a binary variable, defined as a current or non-current smoker or user of any tobacco products at the time of data collection. COVID-19 infection was defined as a positive COVID-19 test before the initiation of COVID-19 vaccination. Self-report data was used to define COVID-19 infection in the majority of cohorts. However, the Electronic Communication of Surveillance in Scotland (ECOSS) COVID-19 testing data was used in the CovidLife cohort, and the E-Health Record registry data was used in EstBB-EHR. Physical comorbidity status was defined as the presence of at least two physical health conditions (hypertension, heart disease, lung disease, chronic renal failure, cancer, diabetes, or immunological conditions), using either self-report data from the cohort-specific questionnaires or EHR data (Supplementary Table 3). Neither data on smoking nor physical comorbidity status were available in the MAP-19 cohort. Further details regarding the time at which variables were defined in each cohort are displayed in Supplementary Table 4.

**Statistical analysis.** Covariates were summarised, stratified by the diagnosis of any mental illness, using mean (standard deviation [SD]) or frequency (percentage), as appropriate. First, multivariable modified Poisson regression models were conducted separately in each cohort study using a standardised analysis protocol. Hereby, models were run for each exposure-outcome combination, to assess the prevalence ratio (PR) with 95% confidence intervals (CI), adjusted for age, sex, previous COVID-19 infection, smoking, and physical comorbidity status. Due to data inavailability in MAP-19, models from this cohort were only adjusted for age, sex, and previous COVID-19 infection. As EstBB-EHR had a high proportion of missing COVID-19 testing data, this cohort used an additional missing indicator for the COVID-19 infection covariate in addition to negative and positive groups, to

avoid dropping large numbers of participants. Subsequently, random effects meta-analyses were performed to aggregate the results from each participating cohort. Heterogeneity was assessed using the I² statistic. Analyses were also performed after stratification by sex.

Three sensitivity analyses were conducted[1]: to explore potential differences related to self-report vs. EHR-based variable definitions, meta-analyses for the primary exposure variable (diagnosis of any mental illness) were run excluding cohorts that used EHR to define exposure and/or outcome variables (i.e., CovidLife, EstBB-C19, and EstBB-EHR)[2], to further explore the potential impact of physical health status, meta-analyses for the primary exposure variable were conducted excluding participants with any chronic physical conditions (using all cohorts except MAP-19)[3], to explore potential differences related to national COVID-19 mitigation strategies and vaccination policies subgroup meta-analyses of Nordic vs. non-Nordic countries were conducted, categorised based on the average of the Oxford COVID-19 Government Response Tracker (OxCGRT) Containment and Health Index for each country between January 2020-September 2021 (Supplementary Table 5)[28]. Statistical analyses were conducted using STATA (version 17.0) and the metafor package in R (version 4.3.0)[29].

### Swedish register study analysis

**Study population.** To gain greater understanding of the role of mental illness type, severity and medication status on the association between mental illness and COVID-19 vaccination uptake, further analyses were conducted using Swedish register data within the SCIFI-PEARL (Swedish COVID-19 Investigation for Future Insights – a Population Epidemiology Approach using Register Linkage) project[30]. Ethical approval was obtained from the Swedish Ethical Review Authority (2020-01800 with subsequent amendments). Informed consent was not required for the Swedish register-based analysis. SCIFI-PEARL is a regularly updated, nationwide register-based study, with individual linkages to multiple registers performed using the unique Swedish Personal Identity Number (PIN). The present study included all individuals aged ≥18 years who were living in Sweden on 27th December 2020 (date of first COVID-19 vaccination in Sweden[31]) and did not die or emigrate on or before the final study end point (30th November 2021).

**Exposure variables.** Recent mental illness was first defined using secondary care-based specialist diagnoses (as denoted by ≥1 ICD-10 code listed in Supplementary Table 6) from all inpatient and outpatient hospital encounters reported in the National Patient Register (NPR) between 1st January 2018 and 26th December 2020. We included the following types of mental illness, both collectively (combined exposure: any mental illness) and as separate exposure variables: substance use disorder [excluding alcohol and tobacco use disorders], alcohol use disorder, tobacco use disorder, psychotic disorders, bipolar disorder, depression, anxiety, and stress-related disorders. The NPR includes data on specialist care only, whereas patients with milder mental illness are often treated in primary care. Therefore, we also identified information on recent prescribed use of the following psychiatric medications (collectively termed: any psychiatric medication): antidepressants, anxiolytics, hypnotics/sedatives, and antipsychotics (defined by ≥1 Anatomical Therapeutic Chemical (ATC) codes displayed in Supplementary Table 7) between 1st January 2018 and 26th December 2020, according to the National Prescribed Drug Register (NPDR). Prescribed use of psychiatric medication was used both to ascertain mental illnesses not attended to by specialist care and as a proxy for treatment of mental illness.

To investigate the effect of disease severity and medication status, an alternative (multi-level) categorisation of exposure variables was used for any mental illness, depression, anxiety, and psychotic disorder, with the following categories[1]: no specialist diagnosis of mental illness and no prescribed use of psychiatric medication (i.e., reference

category)[2], prescribed use of psychiatric medication without specialist diagnosis (i.e., proxy of milder illness with medical treatment)[3], specialist diagnosis with prescribed use of psychiatric medication (i.e., proxy of more severe illness with medical treatment), and[4] specialist diagnosis without prescribed use of psychiatric medication (i.e., proxy of more severe illness without medical treatment).

**Outcome variables.** For the definition of outcome variables, the National Vaccination Register (NVR) was used, which includes information on all COVID-19 vaccinations conducted in Sweden from December 2020[30]. Two outcome variables were used: (i) first dose of a COVID-19 vaccine by 30th September 2021, and (ii) second dose of a COVID-19 vaccine by 30th November 2021. By using this end date for outcome (ii) we ensured that all individuals vaccinated with a first dose up until 30th September would have had the recommended time period between their first and second vaccine dose[32]. Individuals were included in the analysis of outcome (ii) if they had received the first dose of any COVID-19 vaccine, except the JCOVDEN vaccine, by 30th September 2021.

**Covariates.** As in the COVIDMENT study analysis, potential confounders known, based on literature and biological relevance, to be associated with both the exposure (mental illness) and outcome (COVID-19 vaccination uptake) were included as covariates. Covariates included age and sex (identified through the Total Population Register (TPR)), region of residence, highest educational attainment, cohabitation status, and income (identified from the Swedish Longitudinal Integrated Database for Health Insurance and Labour Market Studies (LISA) in 2020), severe COVID-19 infection and the Charlson Comorbidity Index (CCI). Severe COVID-19 infection was defined as an inpatient or ICU visit due to COVID-19 before the study start date (27th December 2020), whilst CCI was calculated at the study start date, using diagnostic data identified from 1st January 2015 onwards in the NPR[33].

**Statistical analysis.** All covariates were summarised, stratified by the diagnosis of any mental illness, using mean (standard deviation [SD]) or frequency (percentage), as appropriate. Multivariable modified Poisson regression models were run for all exposures and outcomes to assesss PR and 95% CIs of COVID-19 vaccination uptake in relation to mental illness, adjusted for the covariates listed above. Stratified analyses were conducted by sex and the presence of chronic physical condition(s) (defined as a CCI score of ≥1). Complete case analysis was used throughout. Statistical analyses were conducted using STATA (version 17.0), and figures were created using R (version 4.3.0)[29].

### Reporting summary
Further information on research design is available in the Nature Portfolio Reporting Summary linked to this article.

## Data availability
The data used in this study are compiled in the COVIDMENT cohorts in Estonia, Iceland, Norway, Scotland, and Sweden (for further information visit www.covidment.is), and any data access is subject to approval of Ethical Review Panels within each country. Interested researchers can obtain access to deidentified data by submitting a proposal to the respective PIs of each cohort (for Estonia Biobank: kelli.lehto@ut.ee, for C-19 Resilience: unnurav@hi.is, for MAP-19: omid.ebrahimi@psy.ox.ac.uk, for MoBa: helga.ask@fhi.no, for CovidLife: Daniel.McCartney@ed.ac.uk, for Omtanke: fang.fang@ki.se) who can assist with submitting an amendment to the ethics review boards. The use of linked Swedish register data for the present research purposes by the study PI and collaborators was approved by the Swedish Ethical Review Authority. Study data from Swedish registers are not available for wider sharing due to policies and regulations in Sweden. Swedish register data are available de novo to all researchers through applications to the respective original register holders, after appropriate ethical approvals have been obtained. For the data used in this study the relevant register holders and contact web sites are: Statistics Sweden (SCB, https://www.scb.se/en/), the National Board of Health and Welfare (Socialstyrelsen, https://www.socialstyrelsen.se/en/), the Public Health Agency of Sweden (Folkhälsomyndigheten, https://www.folkhalsomyndigheten.se/the-public-health-agency-of-sweden/), and the Swedish Intensive Care Registry (SIR) (Svenska Intensivvårdsregistret, https://www.icuregswe.org/en/). The response time for data applications, following the necessary ethical approvals, varies by the demand and capacity at each register. It can range from months to one or even two years, depending on the circumstances. Collaboration using the linked study register data may be possible (subject to adequate/appropriate ethical approvals in place, GDPR provisions and other legal conditions), by contacting the study PI (https://www.gu.se/en/research/scifi-pearl).

## Code availability
Statistical code used for the COVIDMENT and Swedish register analysis is publicly available on GitHub (https://github.com/marymbarker22/covid_19_vaccination_paper).

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

## Acknowledgements

We would like to thank all of the participants who contributed data to the COVIDMENT project. The Norwegian Mother, Father and Child Cohort Study is supported by the Norwegian Ministry of Health and Care Services and the Ministry of Education and Research. We are grateful to all the participating families in Norway who take part in this ongoing cohort study. We thank all the participants of Estonian Biobank for their contribution to this research. We would also like to acknowledge the work of the Estonian Biobank research team: Mait Metspalu, Andres Metspalu, Lili Milani, Tõnu Esko. Estonian Biobank is supported by the European Union through the European Regional Development Fund (https://ec.europa.eu/regional_policy/funding/erdf_en) [Project No. 2014-2020.4.01.15-0012]. Generation Scotland received core support from the Chief Scientist Office of the Scottish Government Health Directorates (https://www.cso.scot.nhs.uk) [CZD/16/6] and the Scottish Funding Council (https://www.sfc.ac.uk) [HR03006] and is currently supported by the Wellcome Trust (https://wellcome.org) [216767/Z/19/Z]. Recruitment to the CovidLife study was facilitated by SHARE- the Scottish Health Research Register and Biobank (https://www.registerforshare.org). SHARE is supported by NHS Research Scotland, the Universities of Scotland and the Chief Scientist Office of the Scottish Government. Figures 1–3, and Supplementary Fig. 2 were created with BioRender.com released under a CC BY-NC-ND license. This work was primarily supported by grants from NordForsk (https://www.nordforsk.org) [COVIDMENT, grant numbers 105668 and 138929] to the respective institutions of UAV, FF, HA, and OAA, and Horizon 2020 [CoMorMent, 847776] to the institutions of OAA, FF and UAV, in addition to FORTE (Swedish Research Council for Health, Working Life and Welfare) [2022-00579 to FF]. KL and KK were supported by the Estonian Research Council (https://etag.ee/en/) [grant PSG615 to KL] and Estonian sub-project of Nord-Forsk project no. 105668 to KL. SCIFI-PEARL has funding to support this study from the SciLifeLab National COVID-19 Research Program (https://www.scilifelab.se/capabilities/pandemic-laboratory-preparedness/pandemic-response/national-program/), financed by the Knut and Alice Wallenberg Foundation (https://kaw.wallenberg.org/en) [grants KAW 2021-0010/VC2021.0018 (to FN: PI) and KAW 2020.0299/VC 2022.0008 (to FN: PI)], and the Swedish Research Council (https://www.vr.se/english.html) [grants 2021-05045 (FN: co-applicant) and 2021-05450 (FN: co-applicant)]. SCIFI-PEARL also has basic funding based on grants from the Swedish state under the agreement between the Swedish government and the county councils, the ALF agreement (https://www.vr.se/english/mandates/clinical-research/clinical-research-in-the-alf-regions.html) [grants ALFGBG-938453 (to FN: P.I.), ALFGBG-971130 (to FN: P.I.), ALFGBG-978954 (to FN: P.I.] and previously from a joint grant from FORTE and FORMAS (Research Council for Environment, Agricultural Sciences and Spatial Planning (https://formas.se/en/start-page.html) [grant 2020-02828 (to FN: P.I.)]. OAA and BW were also supported by the Research Council of Norway (https://www.forskningsradet.no/en/) [#223273 to OAA and #324620 to HA, respectively]. The funders had no role in study design, data collection and analysis, decision to publish, or preparation of the manuscript.

## Author contributions

The participating COVIDMENT cohorts and/or their data collections were designed by I.M., A.L., A.K.K., E.E.J., A.Ho., S.U.J., H.A., D.L.M., O.V.E., K.L., U.A.V., F.F. and their respective teams. H.L. and F.N. provided the registry-based data, server and software to conduct the analysis. M.M.B., F.F., and U.A.V. directed the combined effort of this study implementation. M.M.B., E.H., H.L., U.A.V., F.N. and F.F. designed the analytical strategy in close collaboration with all team members and all authors helped to interpret the findings. M.M.B. and G.M. conducted the literature review and M.M.B., K.K., I.M., H.M., B.W. and X.D. drafted the manuscript under the supervision of U.A.V., F.N. and F.F.. H.A., D.L.M.,

O.V.E., K.L., U.A.V., F.N. and F.F. contributed equally as senior authors. All authors (M.M.B., K.K., I.M., H.M., B.W., X.D., G.M., E.H., E.U.G., H.L., A.L., A.K.K., A.C., M.F., A.H., E.E.J., E.B.T., E.M.F., A.Ho., R.M., G.T., K.Á., J.J., O.A.A., P.F.S., S.U.J., T.A., R.E.B., H.A., D.L.M., O.V.E., K.L., U.A.V., F.N., F.F.) revised the manuscript for critical content and approved the final version of the manuscript. M.M.B., G.M. and H.L. had access to and verified the data.

## Funding

## Competing interests

EMF received speaker's honoraria from Astra Zeneca. OAA is a consultant for Cortechs.ai, and received speaker's honoraria from Janssen, Lundbeck, Sunovion, Otsuka. PFS is a shareholder and on the advisory committee of Neumora Therapeutics. FN owns some AstraZeneca shares. The remaining authors declare no competing interests.

## Additional information

[1]Institute of Environmental Medicine, Karolinska Institutet, Stockholm, Sweden. [2]Estonian Genome Centre, Institute of Genomics, University of Tartu, Tartu, Estonia. [3]Centre of Public Health Sciences, Faculty of Medicine, School of Health Sciences, University of Iceland, Reykjavik, Iceland. [4]Centre for Genomic and Experimental Medicine, Institute of Genetics & Cancer, University of Edinburgh, Edinburgh, United Kingdom. [5]PsychGen Centre for Genetic Epidemiology and Mental Health, Norwegian Institute of Public Health, Oslo, Norway. [6]Department of Child Health and Development, Norwegian Institute of Public Health, Oslo, Norway. [7]Department of Psychology, University of Oslo, Oslo, Norway. [8]Modum Bad Psychiatric Hospital and Research Center, Oslo, Vikersund, Norway. [9]School of Public Health and Community Medicine, Institute of Medicine, Sahlgrenska Academy, University of Gothenburg, Gothenburg, Sweden. [10]Institute of Psychology, Leiden University, Leiden, The Netherlands. [11]Department of Medical Epidemiology and Biostatistics, Karolinska Institutet, Stockholm, Sweden. [12]Department of Rheumatology, University Hospital, Reykjavik, Iceland. [13]NORMENT Centre, Division of Mental Health and Addiction, University of Oslo, Oslo, Norway. [14]Oslo University Hospital, Oslo, Norway. [15]Department of Genetics and Psychiatry, University of North Carolina at Chapel Hill, Chapel Hill, NC, USA. [16]The Norwegian Mother, Father and Child Cohort Study, Norwegian Institute of Public Health, Oslo, Norway. [17]PROMENTA Research Center, Department of Psychology, University of Oslo, Oslo, Norway. [18]Department of Experimental Psychology, University of Oxford, Oxford, United Kingdom. [19]Department of Epidemiology, Harvard T.H. Chan School of Public Health, Harvard University, Boston, MA, USA. [20]These authors contributed equally: Mary M. Barker, Kadri Kõiv, Ingibjörg Magnúsdóttir, Hannah Milbourn, Bin Wang, Xinkai Du. ✉e-mail: mary.barker@ki.se

