## [Peer Review File · Nature Communications]

Mental illness and COVID-19 vaccination: a multinational investigation of observational & register-based dataREVIEWER COMMENTS

Reviewer #1 (Remarks to the Author):

This paper provided a comprehensive analysis across five countries to examine the extent the association between mental illness and COVID-19 vaccination uptake. Overall, the large amount of data was impressive, the methodology was appropriate, and the manuscript was well-written. A few major comments for the authors to consider:

1. It is not surprising with so many large studies, there is heterogeneity between studies. But one key difference is in the exposure variable (mental illness). Some studies used different self-report screeners and the Swedish register study used provider diagnosed disorders. How was this managed and can this heterogeneity be further explored?
2. I may have missed it, but since there are various cultural factors and sociopolitical forces at play around COVID-19 vaccination specific to each country, did the authors examine the association between mental disorder and vaccination WITHIN each country first before combining the data from all countries for cross-country analyses? I think it may be useful to examine whether this was first replicated within each country first, or perhaps it wasn't?
3. There are various important country-level and individual-level variables that were not included and perhaps may not have been feasible to collect but nonetheless may have influenced results. Country-level variables might include the way the COVID-19 vaccination was rolled out, what mandates were attached, and how they were prioritized for different groups. Individual-level variables (such as in Table 1) were missing key, potentially influential variables, like income level, engagement with healthcare services, and COVID-19 infection or symptoms.

Reviewer #2 (Remarks to the Author):

This study examines the association between mental illness and COVID-19 vaccination coverage in a multinational study and a Swedish register-based study. The study suggests that the vaccination coverage in general is high. In the multinational part of the study, no association was found. In the Swedish study, an association between having a diagnosis and not receiving medication had 9% lower vaccination uptake.

Major

Abstract:

Why five countries? It is not very meaningful for the reader as the background/aim. Why not ten countries for instance. It seems random.

COVIDMENT cohort needs to be described/introduced in some way.

Findings

The psychiatric medication has not been introduced in the methods.

The presentation of the result of psychiatric disorder lacks some context to be meaningful.

I suggest the authors to present the estimate for the second dose of vaccine.

Please specify comparison groups.

Conclusions

When concluding on types of mental illness, the authors should present and describe these data in Methods and Results.

The authors don't know why people have been successful with the vaccine. Based on this study there is not a link between the vaccination campaign and vaccination rate as this has not been studied.

According to psychiatric medication there seems to be an assumption that all people with mental illness should be in psychiatric medication. Is that correctly assumed?

I think the conclusions need to be revised to be clearer.

Introduction

The authors could consider to look into the large Danish study of COVID-19, morbidity and mortality which addresses severe mental illness, psychiatric hospitalization, substance use disorders, homelessness ect. I think it could contribute to the introduction section regarding the high-risk populations. This study has not been included in the umbrella review. It also offers data on testing.

<https://doi.org/10.1016/j.lanepi.2022.100421>

When referring to specific studies on vaccination coverage the authors refer to mental illness discussing inconsistent results. However, the authors needs to differentiate between mental illness and severe mental illness, psychiatric admission ect. It is not possible to compare the results across different definitions of mental illness if some use severe mental illness and other any mental illness. This needs to be specified. Also according to the type of cohorts in these studies. The study with no significant differences was based on veterans. Is that directly comparable to the other cohorts compared to? This should be clearer for the reader.

Little is known regarding effect of treatment... Do we know something then please specify what is known.

... could differ depending on the type, severity and treatment status... Yes this is correct, but which type of studies will actually be able to say anything about treatment status? I think this is more than just saying whether people are using psychiatric medication.

When introducing using data from seven cohort studies.. the readers have not heard about COVIDMENT. I think these data needs to be presented in the Introduction before the aim. And it is not clear why these data should be combined with Swedish data.

One study could examine the Swedish data and one could examine COVIDMENT, but why these two together? It is not clear.

Treatment status – although the authors describe that it covers prescribed psychiatric medications, I think this should be revised and focus on medication rather than the broader treatment concept. Please specify the hypothesis for this study.

Since most studies referred to in the Introduction focused on severe mental illness, it should be clearer why focus here is on the mental illness in general?

Methods

What is the background for the COVIDMENT Study? Why is these countries databases combined? It needs explanation.

Why was anxiety and depression selected for specific analyses?

Swedish study exposure:

What was the basis for the definition of any mental illness? Does it include substance use disorders? In that case I would suggest specifying this and using another term such as psychiatric disorder including substance use disorders. But it doesn't seem to be including all other mental disorders? Why not? This needs to be specified in the main text and not only in the appendix.

Please describe why 2018-2020 was chosen for exposure.

Line 193 – please remember to refer to codes in appendix.

It is not clear why this study examines more disorders than those selected in the other study.

How were the covariates selected? Why were these factors used for adjustment?

Results

Line 234-237. Please add numbers and % for these results in the text. And add information about infection status.

Table 2+4. Could the authors add the test result for the differences in the table?

Line 257-258: What about the results from Norway on anxiety and depression?

Line 268-269. The sensitivity analyses results need to be reported more clearly saying what they showed just in a few lines.

Line 271-277. Please add some numbers and %.

The subtitles in the Results section is not very helpful as they not clearly show differences in the cohorts.

Line 307-313. I would suggest presenting estimates for some of the findings regarding diagnosis. For instance, psychosis and substance use disorders.

Discussion

How does the results relate to other studies and the previous mentioned studies?

How comparable are these countries.

Line 327. I would remove the implications to the end of the manuscript and instead present some discussion of the results first.

Line 346. Could the authors please elaborate on the differences between the specific countries included.

Line 347-. In my opinion there is too much focus on the strengths of the study and too little on the limitations. Consider excluding the part with the strengths and elaborate on the limitation part.

Line 349. As I understand you don't combine the Swedish data with the COVIDMENT.

Line 355. Generalization is not the limitation. Please elaborate on the limitation part and then discuss generalizability afterwards.

Please elaborate on the selection bias problem. What do you know from the included cohorts? And in what direction would it influence the results?

I think there is a need for a discussion of more limitations to the study. Some of them should be:

- The problems with testing-patterns between different countries and different sub-groups (for instance include the suggested reference above, which shows that some high-risk populations are less likely to be tested)
- Use of different tests PCR, antigen etc. no information on this. Some people will have been tested multiple times, other not at all. What would this limitation mean for the results?
- Could the authors adjust for testing patterns in the analyses from Sweden or present any results on test probability?
- Limitation would also be different adjustments and missing information – this could be a problem when linking data from different countries.
- Problems with the timing of diagnosis
- It's a limitation that there are no results for severe mental illness, which has been used in most other studies. There is a need of discussion of the difference between severe mental illness vs. any mental illness.
- The problems with information on medication and how this relates to treatment – we don't know if people not receiving medication should have had this or not.
- Causality?

It could be made clearer what this study adds to the previous studies.

We would like to thank the reviewers for their insightful comments and feedback. The input and constructive criticism have greatly contributed to improving the quality and clarity of our manuscript. In this response document, we aim to address each comment, suggestion, and query in a comprehensive manner, providing further clarification and additional information where necessary.

Throughout our revised manuscript, we have carefully incorporated the suggested changes, ensuring that the revised version aligns with the journal's guidelines and requirements. Moreover, we have made every effort to address the concerns and provide a robust response that reflects our commitment to producing high-quality research.

In the following sections, we have presented our responses in italics to differentiate them from the comments from the reviewers. Our responses aim to outline the modifications made and provide explanations, where appropriate. We sincerely hope that these responses adequately address the concerns raised and help to further enhance the scientific merit of our manuscript.

Reviewer 1

This paper provided a comprehensive analysis across five countries to examine the extent the association between mental illness and COVID-19 vaccination uptake. Overall, the large amount of data was impressive, the methodology was appropriate, and the manuscript was well-written.

Response:

We thank reviewer 1 for this positive feedback.

A few major comments for the authors to consider:

1. It is not surprising with so many large studies, there is heterogeneity between studies. But one key difference is in the exposure variable (mental illness). Some studies used different self-report screeners and the Swedish register study used provider diagnosed disorders. How was this managed and can this heterogeneity be further explored?

Response:

We agree that the various measurement methods used to define the exposure (mental illness) could result in heterogeneity. With regards to the Swedish register study vs. the COVIDMENT cohort studies, these results are complementary to each other and are presented separately, showing, however, similar conclusions. Among the different COVIDMENT cohort studies included in the meta-analyses, the majority of the cohorts used self-report information to assess the lifetime diagnosis of any mental illness: the PHQ-9 to measure depressive symptoms, and the GAD-7 to measure anxiety symptoms. We presented separately results on self-reported diagnosis of mental illness and self-reported depressive and anxiety symptoms. As the EstBB-EHR used electronic health records to define mental illness diagnosis, we explored in a sensitivity analysis whether the use of self-report v.s. electronic health records affected the association between mental illness and vaccination uptake, by excluding cohorts which used electronic health records to define the exposure and/or outcome. We found no substantial difference in the results for the majority of outcomes (sensitivity analysis described in Methods section [pg 7], with findings described in Results section [pg 11-13]). We have now elaborated on this in the Discussion section, as follows:

“Another consideration is the use of different methods for the identification of exposure (mental illness) or outcome (COVID-19 vaccination uptake) by the COVIDMENT studies, with some using self-report measures and others using electronic health records. However, results from our sensitivity analysis which excluded cohorts that used electronic health records to define the exposures and/or outcomes showed very similar results.” (pg 17-18)

I may have missed it, but since there are various cultural factors and sociopolitical forces at play around COVID-19 vaccination specific to each country, did the authors examine the association between mental disorder and vaccination WITHIN each country first before combining the data from all countries for cross-country analyses? I think it may be useful to examine whether this was first replicated within each country first, or perhaps it wasn't?

Response:

We thank the reviewer for this question, as it has prompted us to make this matter clear in the Methods section. The analyses (multivariate modified Poisson regression models) were first run separately in each country-specific cohort study using a standardised analysis protocol, before the model results were aggregated using random effects meta-analyses. We have now stated the following in the statistical analysis section of the Methods:

“First, multivariable modified Poisson regression models were conducted separately in each cohort study using a standardized analysis protocol. Hereby, models were run for each exposure-outcome combination, to assess the prevalence ratio (PR) with 95% confidence intervals (CI) of vaccination uptake, adjusted for age, sex, previous COVID-19 infection, smoking, and physical comorbidity status...Subsequently, random effects meta-analyses were performed to aggregate the results from each participating cohort.” (pg 6-7)

All of these results are also displayed in the revised Figure 1 (cohort-specific results in addition to the ‘overall’ results from the meta-analyses) of the revised manuscript.

2. There are various important country-level and individual-level variables that were not included and perhaps may not have been feasible to collect but nonetheless may have influenced results. Country-level variables might include the way the COVID-19 vaccination was rolled out, what mandates were attached, and how they were prioritized for different groups. Individual-level variables (such as in Table 1) were missing key, potentially influential variables, like income level, engagement with healthcare services, and COVID-19 infection or symptoms.

Response:

We agree that these variables are important and could be given more attention in the manuscript. Regarding the country-level variables, reviewer 2 has also suggested that we discuss the comparability of the countries. We have therefore now included more discussion of these country-level variables and how they may affect comparability between countries in the Discussion section of the revised manuscript:

“We also observed slight differences between the cohorts, which could have been, at least partially, due to the varying prioritisation schedules for COVID-19 vaccination used in their respective countries. Although international recommendations, such as those from the European Union (EU), suggested vaccine prioritisation for individuals at the highest risk for severe COVID-19, countries could decide which population groups to include in their prioritisation schedules. As a result, some countries, such as

Estonia, did not select individuals with mental illness for priority vaccination whereas other countries, such as Scotland, did.(27-29) Additional country-level variables, such as the specific COVID-19 vaccination roll out processes used, and the relevant mandates, may have also led to between-country differences in vaccination uptake, by facilitating or hindering vaccination uptake in particular population groups.” (pg 18)

Regarding the individual-level variables, some of the included COVIDMENT cohort studies did not collect data on several potential sociodemographic confounders (e.g. income level, educational attainment), therefore we could not adjust for these variables in the analysis of the COVIDMENT cohort studies. However, the Swedish registers contain rich sociodemographic information, and therefore many potential sociodemographic confounders were adjusted for in the Swedish register analysis. COVID-19 infection was included as a confounder in both the COVIDMENT cohort and Swedish register analysis; however, relevant data on engagement with healthcare services was not available for our analyses, and therefore could have resulted in some residual confounding. We have now discussed the adjustment, or lack thereof, for these individual-level variables in the limitations part of the Discussion section:

“Another limitation is the potential for residual confounding. As some of the included COVIDMENT studies did not collect data on several sociodemographic factors such as income and educational level, we were unable to adjust for these potential confounders in the COVIDMENT analysis. However, as rich sociodemographic information is included in the Swedish registers, we were able to adjust for these potential confounders in the register-based analysis, finding similar results to those found in the COVIDMENT cohort analysis, showing that the results were relatively robust even in the presence of these confounders. However, as in all observational studies, there is still the possibility for residual confounding due to factors not adjusted for, one such factor being engagement with healthcare services.” (pg 17)

Reviewer 2

This study examines the association between mental illness and COVID-19 vaccination coverage in a multinational study and a Swedish register-based study. The study suggests that the vaccination coverage in general is high. In the multinational part of the study, no association was found. In the Swedish study, an association between having a diagnosis and not receiving medication had 9% lower vaccination uptake.

Response:

We thank reviewer 2 for their insightful comments.

Major:

Abstract:

- Why five countries? It is not very meaningful for the reader as the background/aim. Why not ten countries for instance. It seems random.

Response:

We have now removed this part of the aim, as we agree that it is not so meaningful for the reader as the aim per se. Instead, we now state that:

“we aimed to investigate the association between mental illness and COVID-19 vaccination uptake, using data from cohort studies and nationwide Swedish registers.”
(pg 2)

- COVIDMENT cohort needs to be described/introduced in some way.

Response:

We thank the reviewer for pointing this out, and have now explained in the Methods section of the Abstract that the seven cohort studies are “included in the multinational COVIDMENT consortium”. In the Findings section we now refer to “data from the COVIDMENT cohort studies” to make it clearer for the reader. (pg 2)

Findings:

- The psychiatric medication has not been introduced in the methods.
- The presentation of the result of psychiatric disorder lacks some context to be meaningful.

Response:

We realise that we previously did not include in the Abstract the definitions of mental illness (both regarding diagnosis and medication use), meaning that the reader lacked context. We have now added the following sentence to the Methods section of the Abstract:

“Mental illness was defined primarily using self-report measures in the COVIDMENT studies, and using clinical diagnosis and prescription data in the Swedish register data.”
(pg 2)

- I suggest the authors to present the estimate for the second dose of vaccine.
- Please specify comparison groups.

Response:

We have now added the estimate for the second dose and specified the comparison group in the Abstract. This sentence now reads:

“In the Swedish register study population, we observed a very small reduction in the uptake of both the first (prevalence ratio [PR]: 0.99, 95% CI: 0.99-0.99, $p < 0.001$) and second (PR: 0.99, 95% CI: 0.99-0.99, $p < 0.001$) dose of a COVID-19 vaccine among individuals with vs. without mental illness; the reduction was however greater among those not using psychiatric medication (PR: 0.91, 95% CI: 0.91-0.91, $p < 0.001$)”. (pg 2)

Conclusions:

- When concluding on types of mental illness, the authors should present and describe these data in Methods and Results.

Response:

We agree that concluding on types of mental illness without describing this in the Methods and Results means that the reader lacks context. Therefore, to avoid the Abstract becoming overly long we have now removed the mention of types of mental illness in the Conclusions.

- The authors don't know why people have been successful with the vaccine. Based on this study there is not a link between the vaccination campaign and vaccination rate as this has not been studied.

Response:

We agree that conclusions regarding the 'vaccination campaign' based on the current study are too speculative, therefore we have removed mentions of the 'vaccination campaign' from the Conclusions.

- According to psychiatric medication there seems to be an assumption that all people with mental illness should be in psychiatric medication. Is that correctly assumed?

Response:

Psychiatric medication was used as a proxy for regular treatment in this study. It is true that some of the individuals with diagnosed mental illness do not need psychiatric medication. However, as the diagnoses identified in this study were from specialist care, it is likely that a substantial proportion of individuals with such diagnoses have relatively severe mental illness and therefore need some treatment. We have discussed this further in the Discussion section of the revised manuscript, stating:

"We used prescribed use of psychiatric medication as a proxy of treatment for mental illness; however, not all individuals with mental illness are in need of treatment with psychiatric medications. As the diagnostic codes used to identify mental illness in this study were from specialist care, we speculated that a substantial proportion of individuals with these diagnostic codes had relatively severe mental illness which would require some kind of treatment. However, further studies should explore patterns of treatment use including psychotherapy and other non-medical treatments." (pg 17-18)

- I think the conclusions need to be revised to be clearer.

Response:

We have now revised the conclusions to make them clearer, based on the comments above. They now read:

"The findings from our study revealed generally high uptake of COVID-19 vaccination among individuals with and without mental illness. However, lower levels of vaccination uptake were noted among subgroups of individuals with unmedicated mental illness which warrants further attention." (pg 2)

Introduction:

- The authors could consider to look into the large Danish study of COVID-19, morbidity and mortality which addresses severe mental illness, psychiatric hospitalization, substance use disorders, homelessness etc. I think it could contribute to the introduction section regarding the high-risk populations. This study has not been included in the umbrella review. It also offers data on testing.

<https://doi.org/10.1016/j.lanepe.2022.100421>

Response:

We thank the reviewer for suggesting this paper, which gives important insights into the risk of severe COVID-19 and COVID-19-related mortality in several high-risk populations, including

those with mental illness. We have now cited this paper, in addition to the umbrella review, in the first paragraph of the Introduction section, stating:

“Furthermore, the risk of severe COVID-19 infection and COVID-19-related mortality has been shown to be significantly higher among certain vulnerable population groups, such as individuals with mental illness (e.g., substance use disorder and psychiatric disorders requiring psychiatric hospital admissions). Therefore, high coverage of COVID-19 vaccination is especially important among these high-risk groups.” (pg 3)

- When referring to specific studies on vaccination coverage the authors refer to mental illness discussing inconsistent results. However, the authors need to differentiate between mental illness and severe mental illness, psychiatric admission ect. It is not possible to compare the results across different definitions of mental illness if some use severe mental illness and other any mental illness. This needs to be specified. Also according to the type of cohorts in these studies. The study with no significant differences was based on veterans. Is that directly comparable to the other cohorts compared to? This should be clearer for the reader.

Response:

Thank you for the good comments. We have edited the second paragraph of the Introduction section (pg 3) to clearly differentiate trends observed between different types of diagnoses/medications. As some studies used the term ‘severe mental illness’ and others compared specific disorders, separating e.g. schizophrenia from depression, we drew comparisons from specific disorders, highlighting that, in general e.g. schizophrenia and substance use disorder were associated with lower vaccination uptake, whereas anxiety and depression (when analysed separately from several mental illness) were associated with higher uptake.

“As such, while the majority of previous studies have demonstrated lower COVID-19 vaccine uptake among individuals with certain types of mental illness such as schizophrenia and substance use disorder, uptake has been shown to be higher among individuals with anxiety or depression, compared to those with no mental illness.(11-14) Furthermore, one study which explored associations between the use of various types of prescribed psychiatric medications and COVID-19 vaccine uptake found that while individuals using antipsychotics, anxiolytics or hypnotics had lower vaccine uptake, no significant difference in uptake was observed for individuals using antidepressants.(15)” (pg 3)

We thank the reviewer for highlighting the problem of comparability between the cohort based on veterans and other studies. As such, we have removed this study, as well as one cohort which only included patients in one region of England, and focused on the previous nationwide studies that are more directly comparable.

- Little is known regarding effect of treatment... Do we know something then please specify what is known.

Response:

We have now removed this sentence for clarity and highlighted that it is ‘medication status’ (i.e. medicated vs. unmedicated mental illness) that has not previously been studied:

“However, previous studies have not investigated the associations between mental illness severity or medication status (i.e. medicated vs. unmedicated mental illness) and COVID-19 vaccination.” (pg 3)

... could differ depending on the type, severity and treatment status... Yes this is correct, but which type of studies will actually be able to say anything about treatment status? I think this is more than just saying whether people are using psychiatric medication.

Response:

We agree that this statement is a bit unclear, so we have now used the term ‘medication status’ rather than treatment status. Please see the manuscript text cited in our response to the comment above.

We have also added the following statement to the Discussion section:

“We used prescribed use of psychiatric medication as a proxy of treatment for mental illness; however, not all individuals with mental illness are in need of treatment with psychiatric medications. As the diagnostic codes used to identify mental illness in this study were from specialist care, we speculated that a substantial proportion of individuals with these diagnostic codes had relatively severe mental illness which would require some kind of treatment. However, further studies should explore patterns of treatment use including psychotherapy and other non-medical treatments.” (pg 17-18)

- When introducing using data from seven cohort studies.. the readers have not heard about COVIDMENT. I think these data needs to be presented in the Introduction before the aim.

Response:

We have now presented COVIDMENT before the study hypothesis, stating that:

“we used data on mental health diagnoses and symptoms from cohort studies included in the multinational COVIDMENT consortium, in addition to diagnostic and prescription data from the nationwide Swedish registers.” (pg 3)

The aim of the COVIDMENT consortium is now provided at the beginning of the Methods section:

“In order to explore the association between mental illness and uptake of COVID-19 vaccination across several countries, we leveraged data from cohort studies included in the COVIDMENT consortium, a multinational research collaboration aimed at studying mental health trajectories associated with COVID-19.” (pg 4)

- And it is not clear why these data should be combined with Swedish data. One study could examine the Swedish data and one could examine COVIDMENT, but why these two together? It is not clear.

Response:

By using both data sources we were able to explore several different phenotypes of mental illness (e.g., different severity and subtypes) in the same study. The analysis of COVIDMENT cohort studies provided data on the lifetime diagnosis of any mental illness (no data on the specific type), as well as anxiety and depressive symptoms. This allowed some differentiation based on severity (diagnosis vs. symptom); however, the role of medication status (as a proxy of severity as well as treatment status) and the effect of different types of mental illness could

not be assessed using the COVIDMENT data. Therefore, we also used the Swedish register data with information indicative of disease severity (specialist care diagnosis versus use of prescribed psychiatric medications alone, i.e., mental illness attended by primary care) and type (diagnoses of different psychiatric disorders or use of different psychiatric medications). Using both data sources therefore allowed us to conduct a study which could give a comprehensive picture of the associations of mental illness diagnoses, medication use, and symptoms with COVID-19 vaccination uptake, which is missing from previous literature. We have now made this clearer in the Introduction section, stating:

“However, previous studies have not investigated the associations between mental illness severity or medication status and COVID-19 vaccination. In order to explore this, we used data on mental health diagnoses and symptoms from cohort studies included in the multinational COVIDMENT consortium(16), in addition to diagnostic and prescription data from the nationwide Swedish registers.” (pg 3)

- Treatment status – although the authors describe that it covers prescribed psychiatric medications, I think this should be revised and focus on medication rather than the broader treatment concept.

Response:

We thank the reviewer for this suggestion and have now changed the terminology to ‘medication status’ throughout the manuscript.

- Please specify the hypothesis for this study.

Response:

We have now specified the following study hypothesis:

“Our hypothesis was that individuals with mental illness would have lower uptake of COVID-19 vaccination in general, and that this association would differ by mental illness type, severity, and medication status.” (pg 3-4)

- Since most studies referred to in the Introduction focused on severe mental illness, it should be clearer why focus here is on the mental illness in general?

Response:

Several of the previous studies have also focused on disorders such as anxiety and depression (separately from severe mental illness), in addition to the use of different types of psychiatric medication. We have now made this clearer in the Introduction, as well as highlighting the lack of data on mental illness severity and medication status (pg 3). Please see our response to your comment above.

Methods

- What is the background for the COVIDMENT Study? Why is these countries databases combined? It needs explanation.

Response:

We have now provided this background and explanation at the beginning of the Methods section, stating:

“In order to explore the association between mental illness and uptake of COVID-19 vaccination across several countries, we leveraged data from cohort studies included in the COVIDMENT consortium, a multinational research collaboration aimed at studying mental health trajectories associated with COVID-19.” (pg 4)

- Why was anxiety and depression selected for specific analyses?

Response:

Anxiety and depressive symptoms were included as exposure variables as they are common mental symptoms and were included in the data collection of all COVIDMENT cohorts. We have now added this explanation to the manuscript, stating:

“All included cohort studies also collected data on anxiety and depressive symptoms, therefore these data were included as secondary exposure variables.” (pg 5)

- Swedish study exposure:
What was the basis for the definition of any mental illness? Does it include substance use disorders? In that case I would suggest specifying this and using another term such as psychiatric disorder including substance use disorders. But it doesn't seem to be including all other mental disorders? Why not? This needs to be specified in the main text and not only in the appendix.

Response:

We thank the reviewer for making this comment and prompting us to clarify this. In the analysis of Swedish registers we included the most common psychiatric disorders, namely anxiety and depression (we also studied these in the analysis of COVIDMENT cohort studies), and stress-related disorders, in addition to conditions for which low COVID-19 vaccination uptake has commonly been reported in previous studies, namely schizophrenia and substance use disorder. We have now also added diagnosis of bipolar disorder in the revised manuscript, as it has been included in some previous studies under the classification of severe mental illness. These diagnoses are the most common indications for the studied psychiatric medication (antidepressants, anxiolytics, hypnotics/sedatives, and antipsychotics). This has allowed us to study the association between the medication status of mental illness and COVID-19 vaccination uptake, which was missing from the previous literature.

In the revised manuscript, we have now clarified the types of mental illness included in our study, and stated that they are collectively termed ‘any mental illness’ in our paper. During the re-analysis of Swedish register data, through adding bipolar disorder as a study exposure, we noted a coding error in the register data, which prompted further quality checks before we re-conducted all analyses. The inclusion of bipolar disorder and the correction of the coding error have resulted in some changes in the prevalence of mental illness diagnoses and model coefficients, which are generally minor and did not affect our conclusions. We apologize for this and have edited the Results and Discussion sections accordingly.

For example, in the Results section:

“Similarly small differences in first dose uptake were shown for most types of mental illness, except for substance use disorder which had the strongest association with lower vaccination uptake (PR: 0.84, 95% CI: 0.84-0.85, $p < 0.001$), and depression (PR: 1.02, 95% CI: 1.02-1.02, $p < 0.001$) and bipolar disorder (PR: 1.04, 95% CI:

1.03-1.04, $p < 0.001$), for which significantly higher vaccination uptake was found” (pg 13-14)

“with a particularly low uptake of vaccination among individuals with psychotic disorder but no medication use (PR: 0.78, 95% CI: 0.76-0.79, $p < 0.001$).” (pg 14)

“This was also observed for all types of mental illness diagnosis (e.g. PR for substance use disorder: 0.94, 95% CI: 0.94-0.95, $p < 0.001$), PR for psychotic disorder: 1.00, 95% CI: 0.99-1.00, $p < 0.001$) and all types of psychiatric medication.” (pg 15)

For example, in the Discussion section:

“We did, however, show that individuals with substance use disorder had approximately 16% lower uptake of COVID-19 vaccination, while individuals with a specialist diagnosis of mental illness without ongoing psychiatric medication (i.e., proxy of more severe illness without medical treatment) had approximately 9% lower uptake.” (pg 15)

- Please describe why 2018-2020 was chosen for exposure.

Response:

This exposure period was used as we believe that the period constitutes ‘recent’ diagnosis/medication use for mental illness, which we think is most relevant to our research question (association with COVID-19 vaccination). We have now specified in the Methods section that we identified ‘recent’ mental illness. Although our register dataset contained diagnostic data from 2015, the medication data was only available from 2018. The choice of exposure period was also made to set the same exposure period for both diagnosis and medications. The sentences now read:

“Recent mental illness was first defined using secondary care-based specialist diagnoses (as denoted by ≥ 1 ICD-10 code listed in Supplementary Table 5) from all inpatient and outpatient hospital encounters reported in the National Patient Register (NPR) between 1st January 2018 and 26th December 2020... Therefore, we also identified information on recent prescribed use of the following psychiatric medications (collectively termed “any psychiatric medication”): antidepressants, anxiolytics, hypnotics/sedatives, and antipsychotics (defined as ≥ 1 Anatomical Therapeutic Chemical (ATC) code displayed in Supplementary Table 6) between 1st January 2018 and 26th December 2020, according to the National Prescribed Drug Register (NPDR).” (pg 8)

- Line 193 – please remember to refer to codes in appendix.

Response:

Thank you for the good suggestion. We have now clarified this (please see citation of revised manuscript text in comment above).

It is not clear why this study examines more disorders than those selected in the other study.

Response:

One major benefit of including the Swedish register-based analysis was that we were able to expand upon the results from the COVIDMENT cohort studies, including more types of psychiatric disorders, and therefore get a more comprehensive picture of the association between mental illness and COVID-19 vaccination. As such, whilst data on mental illness available from all of the included COVIDMENT cohort studies were regarding lifetime

diagnosis of ‘any mental illness’ and measurement of anxiety and depressive symptoms, the register analysis enabled us to differentiate between many different types of psychiatric disorders, in addition to the use of different types of prescribed psychiatric medication.

- How were the covariates selected? Why were these factors used for adjustment?

Response:

Covariates known to be associated with both the exposure (mental illness) and outcome (COVID-19 vaccination), based on literature and biological relevance, were included as potential confounders in the analyses. This explanation has been added to both the COVIDMENT and Swedish register parts of the Methods section:

-COVIDMENT Study Analysis:

“Potential confounders known, based on literature and biological relevance, to be associated with both the exposure (mental illness) and outcome (COVID-19 vaccination uptake) were included as covariates.” (pg 6)

-Swedish Register Study Analysis:

“As in the COVIDMENT study analysis, potential confounders known, based on literature and biological relevance, to be associated with both the exposure (mental illness) and outcome (COVID-19 vaccination uptake) were included as covariates.” (pg 9)

Results

- Line 234-237. Please add numbers and % for these results in the text. And add information about infection status.

Response:

We have now added the numbers and % of the covariate categories (including infection status) in the text. This paragraph now reads:

“The proportion of females was higher among individuals with (72.0%) vs. without (60.9%) a diagnosis of any mental illness, while the mean age was higher among those with (48.5 [SD: 1.8] years) vs. without (47.8 [SD: 3.6] years) such diagnosis (Table 1). The proportion of individuals with a previous COVID-19 infection was similar between the two groups (2.5% and 2.3% respectively), while the proportions of individuals who smoked or had ≥ 1 chronic physical condition were higher among those with (21.5% smoked, 66.9% had ≥ 1 chronic physical condition) vs. without (17.0% smoked, 36.8% had ≥ 1 chronic physical condition) a diagnosis of any mental illness. Low levels of missing data were observed for the majority of covariates.” (pg 10-11)

- Table 2+4. Could the authors add the test result for the differences in the table?

Response:

The results of the main statistical tests conducted (multivariate Poisson regression models), investigating the uptake of vaccination by exposure status, are all presented in Figures 1-3. We think it is clearer to present them as figures, including the results for all outcomes, rather than duplicating the results in the tables. Statistical tests were not conducted for Tables 1 and 3. Instead, the tables were used to merely show the distribution of study covariates by exposure status, as is recommended to avoid the Table 1 fallacy in epidemiological studies.

- Line 257-258: What about the results from Norway on anxiety and depression?

Response:

We thank the reviewer for this point, and agree that that the Norwegian anxiety and depression results should be described. Therefore, we have now added the following text to the manuscript:

“No association was observed between anxiety or depressive symptoms and vaccination uptake in the overall study population or among males or females separately. However, results from the MoBa cohort showed that uptake was slightly lower among individuals with vs. without anxiety (PR: 0.97, 95% CI: 0.96-0.99) or depressive (PR: 0.98, 95% CI: 0.97-0.99) symptoms.” (pg 12)

- Line 268-269. The sensitivity analyses results need to be reported more clearly saying what they showed just in a few lines.

Response:

We have now reported the results of the sensitivity analyses separately for each vaccination outcome, to add clarity – e.g.

“Results from sensitivity analyses which excluded cohorts using electronic health records for the definition of exposure and/or outcome, or excluded individuals with any chronic physical condition, also showed no significant difference in uptake of the first dose by 30th September 2021 between those with vs. without a diagnosis of any mental illness (Supplementary Table 10).” (pg 11)

- Line 271-277. Please add some numbers and %.

Response:

We have now added some numbers and % to describe the distribution of covariates. This paragraph now reads:

“Among the 8,080,234 individuals included in the Swedish register study population, individuals with a diagnosis of any mental illness were more likely to be female (55.7% vs. 49.6%) and younger (45.2 years vs. 50.2 years), compared to those without such diagnosis (Table 3). Individuals with a mental illness diagnosis had a lower prevalence of university education (28.7% vs. 38.5%), were less likely cohabiting (22.3% vs. 42.5%) or in the highest quartile of income (13.0% vs. 25.8%), and had a higher prevalence of chronic physical conditions, as denoted by a CCI of ≥ 1 (19.0% vs. 11.0%). Although the prevalence of severe COVID-19 infection was low in the overall study population (0.4%), the prevalence was higher among individuals with (0.8%) vs. without (0.3%) any mental illness diagnosis. The proportion of missing data for all covariates was low ($\leq 2.5\%$).” (pg 13)

- The subtitles in the Results section is not very helpful as they not clearly show differences in the cohorts.

Response:

We have now removed sub-headings for each vaccination outcome in the Results section, and kept only those referring to the study population being discussed – i.e. “COVIDMENT Study Analysis” and “Swedish Register Study Analysis” to help guide the reader.

- Line 307-313. I would suggest presenting estimates for some of the findings regarding diagnosis. For instance, psychosis and substance use disorders.

Response:

We have now added estimates for substance use disorder and psychotic disorder. The text now reads:

“This was also observed for all types of mental illness diagnosis (e.g. PR for substance use disorder: 0.94, 95% CI: 0.94-0.95, $p < 0.001$), PR for psychotic disorder: 1.00, 95% CI: 0.99-1.00, $p < 0.001$) and all types of psychiatric medication.” (pg 15).

Discussion

- How does the results relate to other studies and the previous mentioned studies?

Response:

We thank the reviewer for this comment. This part of the discussion had been minimised to keep the word count down in the original manuscript; however, we agree that this is important to discuss and have added an additional paragraph (paragraph 2) in the Discussion section to put our results into the context of previous studies. This paragraph reads:

“Our findings support the results of existing nationwide studies. Accordingly, previous studies have also shown significantly lower uptake of COVID-19 vaccination among individuals with substance use disorders.(11, 13) Some previous studies have shown a higher uptake of COVID-19 vaccination among individuals with anxiety or depression,(13, 14) while another revealed an association between the use of anxiolytics, but not antidepressants, and lower COVID-19 vaccination uptake.(15) The findings from our study similarly found conflicting results, namely that results from the COVIDMENT study population generally showed no association between anxiety or depressive symptoms and COVID-19 vaccination, while results from the Swedish register population showed a higher first dose uptake among individuals with a specialist diagnosis of depression, but not anxiety. Furthermore, our findings show that individuals with a diagnosis of depression but not using prescribed medication had a lower uptake of the first dose of a COVID-19 vaccine. These results indicate that the association between anxiety or depression and COVID-19 vaccination uptake may differ by severity and medication status.” (pg 15-16)

- How comparable are these countries.

Response:

We thank the reviewer for bringing up this point, and it relates to a comment from reviewer 1, who wrote: “There are various important country-level and individual-level variables that were not included and perhaps may not have been feasible to collect but nonetheless may have influenced results. Country-level variables might include the way the COVID-19 vaccination was rolled out, what mandates were attached, and how they were prioritized for different groups.”

We have now included more discussion about these factors in the Discussion section, elaborating on the prioritisation of mental illness (or not) in specific countries included in our study and stating that:

“Additional country-level variables, such as the specific COVID-19 vaccination roll out processes used, and the relevant mandates, may have also led to between-country differences in vaccination uptake, by facilitating or hindering vaccination uptake in particular population groups.” (pg 18)

- Line 327. I would remove the implications to the end of the manuscript and instead present some discussion of the results first.

Response:

We thank the reviewer for this suggestion, and have now moved the implications to allow for the discussion of the results, and the strengths and limitations of the study, to be presented first.

- Line 346. Could the authors please elaborate on the differences between the specific countries included.

Response:

We have now provided country-specific examples from Estonia and Scotland, stating:

“Although international recommendations, such as those from the European Union (EU), suggested vaccine prioritisation for individuals at the highest risk for severe COVID-19, countries could decide which population groups to include in their prioritisation schedules. As a result, some countries, such as Estonia, did not select individuals with mental illness for priority vaccination whereas other countries, such as Scotland, did.(27-29)” (pg 18)

- Line 347-. In my opinion there is too much focus on the strengths of the study and too little on the limitations. Consider excluding the part with the strengths and elaborate on the limitation part.

Response:

We think it is important to highlight the key strengths of the study, namely the use of the two complementary data sources and the study’s multinational nature. Regardless, in the revised manuscript we have shortened the strengths section of the Discussion, and expanded the limitations section, including several of the limitations suggested in your comments below (pg 16-18). Please see our responses to your following comments.

- Line 349. As I understand you don’t combine the Swedish data with the COVIDMENT.

Response:

The reviewer is correct that the COVIDMENT and register data were not combined. This sentence is meant to imply that using both data sources separately combined the benefits of each. We have now re-worded the sentence to avoid misunderstanding, and it now reads:

“Firstly, the complementary use of the two different types of prospectively collected data sources allowed for the benefits of both the rich self-reported COVIDMENT data and the Swedish national registers, minimising concerns of selection bias, and allowing for the investigation of associations between the severity and medication status (i.e. medicated vs. unmedicated mental illness) of mental illness and COVID-19 vaccination, which has not been possible in previous studies.” (pg 16)

- Line 355. Generalization is not the limitation. Please elaborate on the limitation part and then discuss generalizability afterwards.

Response:

We have now moved the discussion of generalisability (including a discussion of comparability issues between the countries) to a separate paragraph, after the limitation section:

“Although the multinational nature of the study increases the representativeness of the findings, all participating countries have established welfare systems and generally accessible healthcare, meaning that caution should be taken when generalising the results to other global regions. We also observed slight differences between the cohorts, which could have been, at least partially, due to the varying prioritisation schedules for COVID-19 vaccination used in their respective countries. Although international recommendations, such as those from the European Union (EU), suggested vaccine prioritisation for individuals at the highest risk for severe COVID-19, countries could decide which population groups to include in their prioritisation schedules. As a result, some countries, such as Estonia, did not select individuals with mental illness for priority vaccination whereas other countries, such as Scotland, did. (27-29) Additional country-level variables, such as the specific COVID-19 vaccination roll out processes used, and the relevant mandates, may have also led to between-country differences in vaccination uptake, by facilitating or hindering vaccination uptake in particular population groups.” (pg 18)

- Please elaborate on the selection bias problem. What do you know from the included cohorts? And in what direction would it influence the results?

Response:

We have now included more discussion of selection bias. Namely, participants in the COVIDMENT cohort studies could have been less likely to have mental illness and more likely to be vaccinated against COVID-19. However, by also using the Swedish register data (which would not have been affected by selection bias in a similar manner, owing to their nationwide coverage), we were able to cross validate the findings from the cohort data to those from the register data. Similar findings from both the COVIDMENT cohort studies and Swedish register studies showed that the results from the former data source were relatively robust, even in the potential presence of selection bias. We have now added a discussion of this in the revised manuscript:

“Firstly, selection bias could have been present in the COVIDMENT cohorts, whereby participants could have been less likely to have mental illness and more likely to be vaccinated against COVID-19. However, by comparing the findings between the COVIDMENT study population and the Swedish register population, which would not have suffered from selection bias, we were able to investigate whether the potential selection bias in the COVIDMENT cohorts had a substantial effect on the study results. Similar findings from both data sources highlighted the robustness of the study results, even in the potential presence of selection bias.” (pg 16-17)

- I think there is a need for a discussion of more limitations to the study. Some of them should be:
 - The problems with testing-patterns between different countries and different sub-groups (for instance include the suggested reference above, which shows that some high-risk

populations are less likely to be tested)

- Use of different tests PCR, antigen etc. no information on this. Some people will have been tested multiple times, other not at all. What would this limitation mean for the results?
- Could the authors adjust for testing patterns in the analyses from Sweden or present any results on test probability?

Response:

We thank the reviewer for suggesting these points; however, we do not see these as limitations of the present study investigating the association between mental illness and COVID-19 vaccination uptake. However, they do represent interesting research questions on their own for future research.

- Limitation would also be different adjustments and missing information – this could be a problem when linking data from different countries.

Response:

We agree and have now added discussion on this limitation, stating:

“Another potential limitation is the different multivariable adjustments used in the country-specific analysis, due to data unavailability in some cohorts. This, in addition to the presence of differing levels of missing data, may have influenced the pooling of country-specific results and increased the heterogeneity in the meta-analyses.” (pg 17)

- Problems with the timing of diagnosis

Response:

We agree that diagnostic delay could be a limitation of our study, as in all studies using clinical diagnosis to ascertain mental illness. Therefore we have now added the following statement in the Discussion section:

“Additionally, identification of mental illness in the Swedish register data was based on specialist diagnosis or prescribed use of psychiatric medication, which may occur some time after the onset of symptoms.” (pg 17)

- It’s a limitation that there are no results for severe mental illness, which has been used in most other studies. There is a need of discussion of the difference between severe mental illness vs. any mental illness.

Response:

We thank the reviewer for raising this, however, we do not agree that this is a limitation of our study. Although some previous studies have examined severe mental illness as one category, others have instead provided separate results for individual psychiatric disorders, with some finding differing trends for different types of mental illness (e.g. depression vs. schizophrenia) within the same study. Therefore, by providing separate results for different psychiatric disorders we are able to give a more comprehensive picture of the link between mental illness and COVID vaccine uptake. Regardless, we have now included more information in the Discussion section, comparing results between different psychiatric disorders, and describing how these disorder-specific results compare to those found in previous studies (pg 15-16). Please see our response to your comments above.

- The problems with information on medication and how this relates to treatment – we don't know if people not receiving medication should have had this or not.

Response:

We agree and have now added this to the limitations section of Discussion, stating:

“Furthermore, in our multi-level exposure analysis we used diagnosis and prescription data to give indications of the severity level and medication status of mental illness. We used prescribed use of psychiatric medication as a proxy of treatment for mental illness; however, not all individuals with mental illness are in need of treatment with psychiatric medications. As the diagnostic codes used to identify mental illness in this study were from specialist care, we speculated that a substantial proportion of individuals with these diagnostic codes had relatively severe mental illness which would require some kind of treatment. However, further studies should explore patterns of treatment use including psychotherapy and other non-medical treatments.” (pg 17-18)

- Causality?

Response:

We have now added this limitation to the Discussion section, stating:

“Lastly, the nature of this observational study means that it is not possible to ascertain causality based on the findings.” (pg 18)

It could be made clearer what this study adds to the previous studies.

Response:

We have now elaborated on this in the following sections of the manuscript:

-Introduction section:

“However, previous studies have not investigated the associations between mental illness severity or medication status (i.e. medicated vs. unmedicated mental illness) and COVID-19 vaccination. In order to explore this, we used data on mental health diagnoses and symptoms from cohort studies included in the multinational COVIDMENT consortium, in addition to diagnostic and prescription data from the nationwide Swedish registers.” (pg 3)

-Discussion section (strengths paragraph):

“Firstly, the complementary use of the two different types of prospectively collected data sources allowed for the benefits of both the rich self-reported COVIDMENT data and the Swedish national registers, minimising concerns of selection bias, and allowing for the investigation of associations between the severity and medication status (i.e. medicated vs. unmedicated mental illness) of mental illness and COVID-19 vaccination, which has not been possible in previous studies. The multinational nature of our study also enabled us to investigate whether the results found in previous country-specific studies translated to a multinational context.” (pg 16)

REVIEWERS' COMMENTS

Reviewer #1 (Remarks to the Author):

I appreciate the authors work and the opportunity to re-review the revised manuscript. I think the authors have mostly addressed my concerns. I do continue to feel there are major differences between countries, which the authors now discuss further in the Discussion. But they did not attempt to control for any in the analyses. For example, even a dichotomy or three-level categorization of countries that implemented strict and lengthy procedures for social distancing vs. lax or short social distancing measures could be informative or as least important to control for in analyses. The same is true regarding how organized COVID-19 vaccination distribution was and the level of vaccine uptake by country.

Reviewer #2 (Remarks to the Author):

Thank you for the revised manuscript, which has highly improved.

Manuscript reference number: NCOMMS-24-14891A

We would like to thank the reviewers for their further comments and feedback on our revised manuscript. In this response document, we aim to address these comments, providing the details and findings of further analyses, where necessary. We have indicated all text changes made, along with the page numbers at which the changes can be found. We sincerely hope that these responses adequately address the remaining concerns raised.

Reviewer #1 (Remarks to the Author):

I appreciate the authors work and the opportunity to re-review the revised manuscript. I think the authors have mostly addressed my concerns. I do continue to feel there are major differences between countries, which the authors now discuss further in the Discussion. But they did not attempt to control for any in the analyses. For example, even a dichotomy or three-level categorization of countries that implemented strict and lengthy procedures for social distancing vs. lax or short social distancing measures could be informative or as least important to control for in analyses. The same is true regarding how organized COVID-19 vaccination distribution was and the level of vaccine uptake by country.

Response:

We thank the reviewer for their positive feedback regarding the revised manuscript. As suggested by the reviewer, we have now conducted a further sensitivity analysis to take into account country-specific mitigating strategies and vaccination organization. Hereby, we used the containment and health index (CHI), as defined using The Oxford COVID-19 Government Response Tracker (OxCGRT) (<https://www.bsg.ox.ac.uk/research/covid-19-government-response-tracker>). The CHI is comprised of 13 COVID-19 policy indicators, including data on social distancing procedures (e.g. stay-at-home requirements, restrictions on internal movements), COVID-19 testing policies, and vaccine rollout policies. Specifically regarding vaccination, the CHI reflects how organised COVID-19 vaccination distribution was, by including criteria such as the availability and prioritisation of vaccines for various population groups and government investment in COVID-19 vaccines. As the CHIs for the included countries varied over time during the COVID-19 pandemic, for our sensitivity analysis we calculated the average CHI for each country between January 2020-September 2021.

Using this data (displayed in Supplementary Table 5), we categorised the included cohorts into those located in Nordic (Sweden, Norway, Island) vs. non-Nordic (Estonia, UK) countries and conducted a subgroup meta-analysis. As the average CHIs in the Nordic countries were relatively similar, while the CHIs in the non-Nordic countries varied substantially (high in UK, low in Estonia) we were able to ascertain whether the heterogeneity noted in the country-specific associations between mental illness and COVID-19 vaccine uptake could be partially explained by differences in national COVID-19 policies (including mitigation strategies and vaccination practice). We have made the following revisions to the manuscript:

-Methods: "Three sensitivity analyses were conducted: ... (3) to explore potential differences related to national COVID-19 mitigation strategies and vaccination policies we conducted subgroup meta-analyses of Nordic vs. non-Nordic countries, categorized based on the average of the Oxford COVID-19 Government Response Tracker (OxCGRT) Containment and Health Index for each country between January 2020-September 2021 (Supplementary Table 5).(28)" (page 16-17)

*-Results (Uptake of the first dose of a COVID-19 vaccine by 30th September 2021):
“Additionally, results from the third sensitivity analysis showed very similar patterns in the Nordic and non-Nordic country groups, with no significant association between a diagnosis of any mental illness and vaccine uptake observed in either group.” (page 5)*

-Results (Uptake of the first dose of a COVID-19 vaccine by 18th February 2022): Although the results for both Nordic (pooled PR: 0.99, 95% CI: 0.99-1.00; I²: 18.2%, p>0.05) and non-Nordic (pooled PR: 0.98, 95% CI: 0.97-0.99, I²: 67.4%, p<0.05) country groups were very similar, the association between a diagnosis of any mental illness and vaccine uptake was only statistically significant in the latter group. (page 6)

-Discussion: Additional country-level variables, such as the specific COVID-19 mitigation and vaccination policies used may have also led to between-country differences in vaccination, by facilitating or hindering vaccine uptake in particular population groups. However, the relatively similar results observed from our sensitivity analysis, which categorised countries based on national COVID-19 mitigation and vaccination policies, suggest that these factors may not have a large impact on the association between mental illness and COVID-19 vaccination. (page 12)

Reviewer #2 (Remarks to the Author):

Thank you for the revised manuscript, which has highly improved.

Response:

We thank the reviewer for their very positive feedback.